# Video-OPD: Efficient Post-Training of Multimodal Large Language Models for Temporal Video Grounding via On-Policy Distillation

**Jiaze Li** [* † 1]  **Hao Yin** [* 1]  **Haoran Xu** [* 2]  **Boshen Xu** [3]  **Wenhui Tan** [3]  **Zewen He** [1]  **Jianzhong Ju** [‡ 1]  **Zhenbo Luo** [1]  **Jian Luan** [1]

## Abstract

Reinforcement learning has emerged as a principled post-training paradigm for Temporal Video Grounding (TVG) due to its on-policy optimization, yet existing GRPO-based methods remain fundamentally constrained by sparse reward signals and substantial computational overhead. We propose Video-OPD, an efficient post-training framework for TVG inspired by recent advances in on-policy distillation. Video-OPD optimizes trajectories sampled directly from the current policy, thereby preserving alignment between training and inference distributions, while a frontier teacher supplies dense, token-level supervision via a reverse KL divergence objective. This formulation preserves the on-policy property critical for mitigating distributional shift, while converting sparse, episode-level feedback into fine-grained, step-wise learning signals. Building on Video-OPD, we introduce Teacher-Validated Disagreement Focusing (TVDF), a lightweight training curriculum that iteratively prioritizes trajectories that are both teacher-reliable and maximally informative for the student, thereby improving training efficiency. Empirical results demonstrate that Video-OPD consistently outperforms GRPO while achieving substantially faster convergence and lower computational cost, establishing on-policy distillation as an effective alternative to conventional reinforcement learning for TVG.

## 1. Introduction

Video understanding is a fundamental objective in multimodal learning. (Gaidon et al., 2013; Darrell & Pentland, 1993) One of its core challenges is Temporal Video Grounding (TVG) (Gao et al., 2017; Zhang et al., 2023), which aims to localize the temporal segments in a video that correspond to natural language queries, such as identifying when two people are engaged in a fight. By explicitly aligning linguistic semantics with fine-grained temporal visual content, TVG enables precise temporal reasoning and semantic interpretation of videos. Consequently, it serves as a foundational component for a wide range of downstream applications, including video retrieval (Laptev & Pérez, 2007), human–computer interaction (Grauman et al., 2022), and embodied AI (Anderson et al., 2018).

Recent studies, such as Time-R1 (Wang et al., 2025) and TVG-R1 (Chen et al., 2025), have investigated Reinforcement Learning (RL) as a post-training paradigm for TVG, demonstrating notable performance improvements. The core strength of RL arises from its on-policy nature: optimization is performed over trajectories sampled from the current policy, thereby mitigating the distribution shift between training and inference states. As a result, the model can explicitly adapt to its own prediction-induced states, alleviating error accumulation and enhancing robustness in long-horizon temporal decision processes.

However, as illustrated in Figure 1, existing RL-based post-training methods remain limited for TVG due to two key factors. First, Group Relative Policy Optimization (GRPO) (Shao et al., 2024) provides only sequence-level supervision, yielding a single, fixed-size reward per episode regardless of trajectory length. This results in a severe credit assignment problem, as the policy receives no information about which intermediate token-level decisions contribute to success or failure. The resulting reward sparsity leads to poor sample efficiency and is especially detrimental for long-horizon tasks such as TVG. Second, GRPO relies on multiple on-policy rollouts to obtain low-variance reward estimates. In video understanding, where each rollout conditions on long visual contexts, the cost of generating trajectories scales prohibitively with the number of rollouts, incurring substantial computational overhead during training.

Motivated by these challenges, we propose Video-OPD, a highly efficient reinforcement learning framework for

---

[1]MiLM Plus, Xiaomi Inc. [2]Zhejiang University [3]Renmin University of China. Correspondence to: Jianzhong Ju <jujianzhong@xiaomi.com>.

*Proceedings of the 43$^{rd}$ International Conference on Machine Learning*, Seoul, South Korea. PMLR 306, 2026. Copyright 2026 by the author(s).

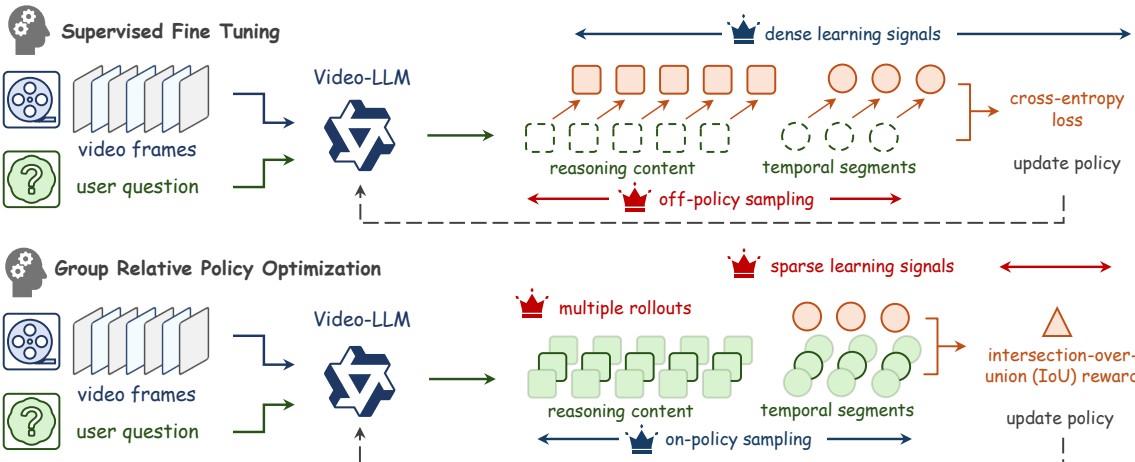

*Figure 1.* Limitations of Supervised Fine-Tuning (SFT) and Group Relative Policy Optimization (GRPO) on Temporal Video Grounding (TVG). **Blue crowns** denote **strengths**, while **red crowns** indicate **weaknesses**. SFT provides dense supervision but is restricted to off-policy optimization, whereas GRPO enables on-policy optimization at the cost of sparse reward signals and multiple rollouts.

TVG that preserves strict on-policy optimization (Lu & Lab, 2025) while providing dense supervisory signals, inspired by recent advances in on-policy distillation. Video-OPD optimizes on trajectories sampled from the current policy, preserving training–inference alignment, while a fixed frontier teacher provides dense, token-level supervision through a reverse KL objective. This design retains the on-policy property critical for mitigating distributional shift, while transforming sparse episode-level rewards into fine-grained, step-wise learning signals. Consequently, each training episode yields substantially richer supervision, enabling more precise credit assignment and significantly reducing gradient variance. Empirically, this dense supervision formulation allows Video-OPD to converge stably and rapidly during training, leading to consistently improved optimization effectiveness. Moreover, Video-OPD eliminates the need for multiple rollouts per training sample when estimating trajectory-level rewards, thereby substantially reducing computational overhead.

Building upon Video-OPD, we introduce Teacher-Validated Disagreement Focusing (TVDF), a lightweight training curriculum that further improves optimization efficiency. TVDF uses ground-truth temporal annotations solely to validate the reliability of the teacher, and prioritizes on-policy trajectories exhibiting large teacher–student disagreement, quantified by aggregated reverse KL divergence. Applied iteratively during training, this curriculum steers Video-OPD toward samples that are both teacher-reliable and maximally informative for the current policy. This indirect yet principled use of annotations enables more effective exploitation of labeled data when direct supervision is suboptimal, resulting in improved sample efficiency and faster convergence.

Comprehensive experiments demonstrate that Video-OPD consistently outperforms GRPO across a wide range of benchmarks. On standard TVG datasets including Charades-TimeLens (Zhang et al., 2025), ActivityNet-TimeLens (Zhang et al., 2025), and QVHighlights-TimeLens (Zhang et al., 2025), Video-OPD achieves an average improvement exceeding 17%, substantially surpassing the roughly 12% gains obtained by GRPO. Beyond TVG tasks, Video-OPD demonstrates strong generalization to broader video understanding benchmarks, including TempCompass (Liu et al., 2024), MVBench (Li et al., 2024), and Video-MME (Fu et al., 2025). Moreover, an analysis of training dynamics shows that Video-OPD converges substantially faster while requiring a markedly lower computational budget, resulting in a more favorable efficiency–performance trade-off.

In summary, this paper presents four key contributions:

- We identify two limitations of GRPO-based post-training for TVG: inefficient optimization due to sparse reward signals, and prohibitive computational overhead induced by long visual contexts and multi-rollout training.

- We propose Video-OPD, an efficient post-training framework for TVG that replaces sparse sequence-level rewards with dense token-level supervision derived from a frontier teacher via a reverse KL objective, enabling more effective credit assignment and faster convergence.

- We introduce TVDF, a lightweight training curriculum that further improves the efficiency of Video-OPD by validating teacher reliability using ground-truth annotations and prioritizing on-policy trajectories with large teacher–student disagreement, allowing more effective use of labeled data without direct supervision.

- Extensive experiments confirm that Video-OPD outperforms GRPO across standard TVG and broader video understanding benchmarks, achieving faster convergence with a reduced computational budget.

## 2. Motivation

In Section 2.1, we first analyze why Supervised Fine-Tuning (SFT) fails to effectively address the challenges of TVG. In Section 2.2, we then examine RL–based post-training methods such as GRPO, showing that while they yield performance gains, they introduce new optimization bottlenecks in the form of sparse sequence-level rewards and costly multi-rollout training procedures. Motivated by these limitations, Section 2.3 distills the core requirements that a principled and effective post-training framework for TVG must satisfy, which directly informs the design of our proposed approach.

### 2.1. Failure of Off-Policy Supervised Fine-Tuning

TVG aims to localize a temporal boundary $[t_s, t_e]$ within a video $v$ that semantically corresponds to a natural-language query $q$. Modern Multimodal Large Language Models (MLLMs) typically formulate TVG as an autoregressive decision process, where the model sequentially predicts temporal actions $a_t \in \mathcal{A}$ conditioned on the state $s_t = (v, q, a_{<t})$, where $\mathcal{A}$ denotes a discrete temporal action space.

As illustrated in Figure 1, SFT often yields suboptimal performance on TVG. A fundamental limitation of SFT lies in its off-policy nature: model parameters are optimized solely on demonstration trajectories sampled from a fixed data distribution, rather than on the state distribution induced by the model's own policy during inference. Formally, SFT minimizes the expected loss over expert trajectories $\tau \sim p_{\text{data}}(\tau)$, while deployment executes trajectories $\tau \sim \pi_\theta(\tau)$, leading to a distributional mismatch between training and inference.

This mismatch results in compounding errors. Once an early prediction deviates from the ground-truth trajectory, the model may enter states that were never observed during training, causing subsequent predictions to progressively deteriorate. Such error accumulation is particularly severe in TVG, where autoregressive temporal decisions are strongly coupled across steps and early localization errors propagate over long horizons, ultimately leading to significant temporal drift and degraded grounding accuracy.

### 2.2. On-Policy Reinforcement Learning Is Not Enough

To overcome the limitations of SFT, recent studies, most notably Time-R1, have investigated reinforcement learning (RL) as a post-training paradigm for TVG and reported consistent performance improvements. Concretely, instead of imitating expert demonstrations as in SFT, Time-R1 directly optimizes a policy $\pi_\theta$ over trajectories $\tau = (a_1, \ldots, a_T)$ sampled from its own induced state distribution $s_t$. Optimization is performed using GRPO, which maximizes an expected trajectory-level objective:

$$\max_\theta \ \mathbb{E}_{\tau \sim \pi_{\theta_{\text{old}}}} \big[ R(\tau) - \beta \, D_{\text{KL}}(\pi_\theta \,\|\, \pi_{\text{ref}}) \big], \quad (1)$$

where $\pi_{\theta_{\text{old}}}$ denotes the policy from the previous optimization step, $\pi_{\text{ref}}$ is a reference policy anchoring the pretrained model, and $\beta$ controls the strength of the KL regularization. The trajectory-level reward $R(\tau)$ is computed using group-relative normalization with importance weighting:

$$R(\tau) = \frac{1}{G} \sum_{i=1}^{G} \frac{\pi_\theta(\tau_i)}{\pi_{\theta_{\text{old}}}(\tau_i)} \cdot \hat{r}_G(\tau_i), \quad (2)$$

where $\{\tau_i\}_{i=1}^{G}$ denotes a group of on-policy trajectories sampled from $\pi_{\theta_{\text{old}}}$, and $\hat{r}_G(\tau_i)$ represents the group-normalized sequence-level reward. In practice, Time-R1 adopts a timestamp-aware Intersection-over-Union (IoU) as the reward function, which augments the standard IoU by penalizing temporal boundary deviations.

By optimizing over trajectories sampled from the current policy, Time-R1 aligns the training state distribution with that encountered at inference time. This on-policy formulation exposes the model to states induced by its own predictions, enabling recovery from early deviations and mitigating the compounding-error problem inherent to off-policy methods. However, while GRPO partially addresses the distributional mismatch of SFT, it introduces two new bottlenecks when applied to TVG, as illustrated in Figure 1.

**Sparse Rewards Lead to Ineffective Credit Assignment.** As formalized in Equation (1), GRPO relies exclusively on trajectory-level supervision. After generating a decision trajectory $\tau = (a_1, \ldots, a_T)$, the model receives a single scalar reward $R(\tau)$ that assesses the overall quality of temporal video grounding. This design yields an extremely sparse learning signal: regardless of the trajectory length $T$, each episode provides only $\mathcal{O}(1)$ feedback. Consequently, the supervision signal does not scale with the temporal depth or structural complexity of the autoregressive process, severely impairing effective credit assignment across individual decisions and undermining stable learning over long horizons.

Concretely, GRPO updates the policy by estimating the trajectory-level policy gradient:

$$\nabla_\theta \mathcal{L}_{\text{GRPO}} = -\mathbb{E}_{\tau \sim \pi_{\theta_{\text{old}}}} \big[ R(\tau) \, \nabla_\theta \log \pi_\theta(\tau) \big]. \quad (3)$$

Notably, the trajectory-level reward $R(\tau)$ is uniformly propagated to all time steps along the trajectory, causing each intermediate token-level decision to receive identical supervision regardless of its actual contribution to the final localization outcome. Consequently, the gradient estimator must implicitly infer responsibility across a long sequence of tightly coupled decisions, leading to ambiguous credit assignment, elevated gradient variance, and slow convergence. We validate this claim through an analysis of GRPO training dynamics (Section 5) and provide a theoretical justification in Section A. This limitation is particularly acute in TVG, where early boundary predictions strongly constrain subsequent actions over extended temporal horizons.

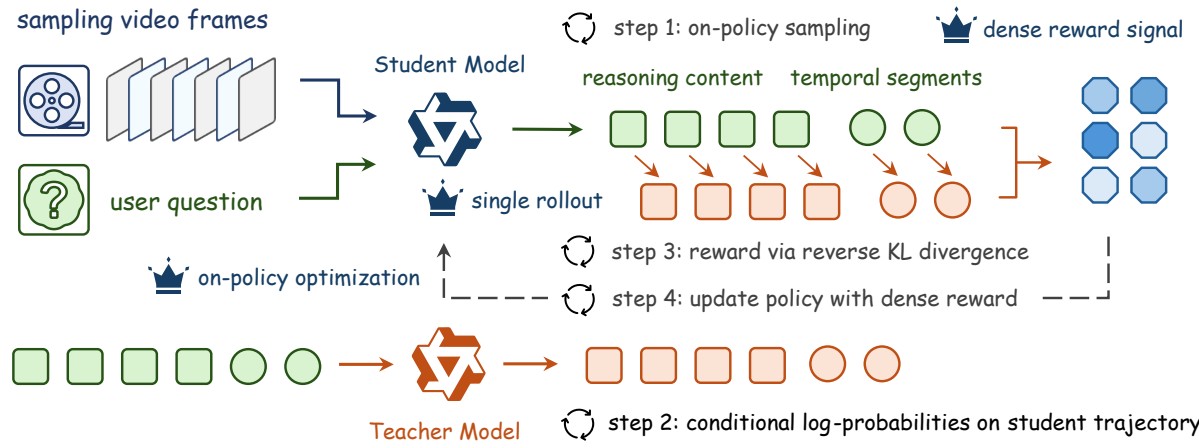

*Figure 2.* Overview of the Video-OPD post-training framework. Video-OPD optimizes trajectories sampled on-policy to maintain training–inference alignment, leverages a fixed frontier teacher to provide dense token-level supervision via reverse KL for fine-grained credit assignment, and eliminates multiple rollouts per sample, substantially reducing computational overhead.

**Multi-Rollout Training Leads to Prohibitive Overhead.**
As shown in Equation (2), GRPO attempts to mitigate the high variance induced by sparse rewards by averaging gradients over multiple rollouts for each training instance. While this strategy can partially stabilize optimization, it incurs a prohibitive computational burden in the context of video understanding. Each rollout requires conditioning on long visual sequences and performing full autoregressive trajectory generation, causing the training cost to scale with both the trajectory length $T$ and the number of rollouts. In practice, this multi-rollout requirement introduces substantial overhead in computation, memory, and latency, particularly for long videos. We validate this claim in Section 5 through an analysis of the training dynamics of GRPO.

### 2.3. Insights for Improved Post-Training Paradigms

SFT provides token-level supervision, yet optimizes the model under a fixed demonstration distribution rather than the on-policy state distribution induced at inference time. This off-policy mismatch leads to compounding errors: early deviations push the model into unseen states without training signal, causing errors to accumulate along the trajectory. In contrast, GRPO optimizes the policy on trajectories sampled from the current policy, aligning training and inference state distributions and mitigating error accumulation in long-horizon decision processes. However, GRPO relies solely on sequence-level supervision, yieldinga single, fixed-size reward per trajectory and providing no explicit signal for attributing credit to intermediate token-level decisions.

Therefore, the proposed post-training paradigm aims to retain the on-policy property, which is critical for mitigating distributional shift, while transforming sparse episode-level rewards into fine-grained, step-wise learning signals. The goal is to enable faster convergence during training and

achieve improved overall training performance. Moreover, we aim to eliminate the need for multiple on-policy rollouts, thereby reducing the computational cost of training.

## 3. Method

In Section 3.1, we introduce Video-OPD, a strictly on-policy reinforcement learning framework for Temporal Video Grounding (TVG) that tightly couples policy optimization with dense, distillation-based supervision. Building on this framework, Section 3.2 presents Teacher-Validated Disagreement Focusing (TVDF), a lightweight training curriculum that uses annotated temporal data exclusively as a validation signal to identify reliable and informative samples, thereby further enhancing optimization efficiency.

### 3.1. On-Policy Distillation for TVG

As illustrated in Figure 2, at each training iteration, Video-OPD performs optimization exclusively on trajectories sampled from the current student policy, while a fixed teacher model is used solely to provide fine-grained, token-level learning signals. This on-policy design preserves distributional alignment between training and inference, while substantially enhancing the efficiency of credit assignment. The overall training pipeline comprises four sequential stages.

**Step 1: On-Policy Trajectory Sampling.** Given a video-query pair $(v, q)$, the student model samples a trajectory

$$\tau = (a_1, \ldots, a_T) \sim \pi_\theta(\cdot \mid v, q), \tag{4}$$

from its current policy. The trajectory consists of both reasoning tokens and the final temporal boundary predictions. During sampling, we record the token-level log-probabilities $\log \pi_\theta(a_t \mid s_t)$ under the student policy, which are subsequently used to construct per-token optimization signals.

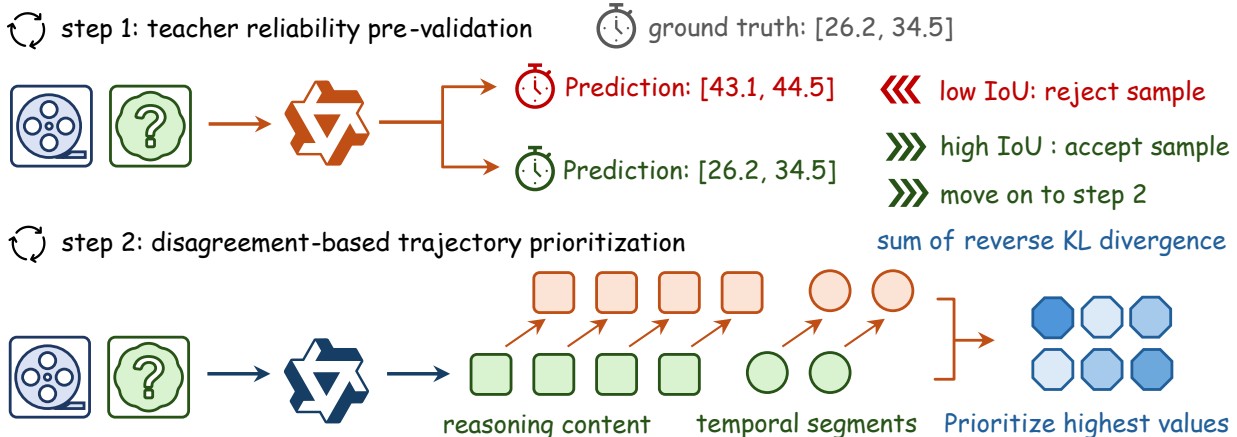

Figure 3. Overview of the Teacher-Validated Disagreement Focusing (TVDF) training curriculum. TVDF iteratively prioritizes trajectories that are both teacher-reliable and maximally informative for the student, thereby improving training efficiency.

By sampling trajectories strictly from the current policy, Video-OPD preserves training-inference alignment and avoids the distributional mismatch and compounding errors observed in off-policy distillation and SFT, particularly in long-horizon video understanding.

**Step 2: Evaluation on Student Trajectory.** A fixed high-capacity teacher $\pi_{\text{tea}}$ is leveraged to obtain the conditional log-probabilities $\log \pi_{\text{tea}}(a_t \mid s_t)$ for each student-generated token. Importantly, the teacher never generates tokens itself and is only used to evaluate the student's on-policy samples, ensuring strict on-policy optimization.

Consequently, the supervision provided by the teacher is aligned with the student's on-policy state distribution, avoiding the distributional mismatch inherent in off-policy distillation methods where teacher distributions are computed by conditioning on ground-truth trajectories.

**Step 3: Dense Token-Level Supervision.** Using the student and teacher log-probabilities, we define a dense, per-token learning signal induced by the reverse KL divergence:

$$r_t = -\Big( \log \pi_\theta(a_t \mid s_t) - \log \pi_{\text{tea}}(a_t \mid s_t) \Big), \quad (5)$$

which captures a pointwise supervisory signal that drives the student policy $\pi_\theta$ to align with the teacher $\pi_{\text{tea}}$ at state $s_t$. Under this formulation, tokens to which the teacher assigns low probability incur proportionally larger penalties, yielding fine-grained, token-level supervision that precisely pinpoints intermediate reasoning steps deviating from the teacher's expected behavior.

By providing supervision at every generation step, Video-OPD transforms sparse, episode-level rewards into dense, fine-grained learning signals, enabling accurate temporal credit assignment across long-horizon reasoning chains. We validate this claim in Section 5 through a comparative analy-

sis of the training dynamics of Video-OPD and GRPO, and provide a theoretical justification in Section A.

**Step 4: Policy Update with Dense Rewards.** The student policy is optimized via standard policy-gradient updates, treating the teacher-derived signal $r_t$ as a token-level reward. Specifically, policy parameters $\theta$ are updated according to

$$-\mathbb{E}_{\tau \sim \pi_{\theta_{\text{old}}}} \left[ \sum_{t=1}^{T} r_t \frac{\pi_\theta(a_t \mid s_t)}{\pi_{\theta_{\text{old}}}(a_t \mid s_t)} \nabla_\theta \log \pi_\theta(a_t \mid s_t) \right], \quad (6)$$

where each action $a_t$ is explicitly supervised by by a step-specific reward $r_t$. Unlike GRPO, which relies on sparse episode-level rewards, this dense token-level supervision enables fine-grained credit assignment across individual decisions and substantially reduces gradient variance, leading to more stable and efficient optimization.

Moreover, because each trajectory yields dense, token-level supervision, Video-OPD requires only a single rollout per training sample. This obviates the need for grouped rollouts or reward normalization strategies commonly used in GRPO, significantly reducing computational overhead. We validate this claim in Section 5 through a comparative analysis of the training dynamics of Video-OPD and GRPO.

### 3.2. Teacher-Validated Disagreement Focusing

We introduce Teacher-Validated Disagreement Focusing (TVDF), a training curriculum designed to further enhance optimization efficiency. The key observation motivating TVDF is that Video-OPD performs policy optimization without relying on annotated temporal video grounding data. Nevertheless, such annotations encode valuable information that can be exploited during training. TVDF therefore incorporates ground-truth temporal annotations solely as a validation signal, rather than as a direct supervision target.

As illustrated in Figure 3, TVDF consists of two comple-

*Table 1.* Evaluation of Video-OPD on three TVG benchmarks. $^\star$ denotes our reproduced results. **Bold** values indicate the best performance.

| Models for Evaluation | Charades-TimeLens | | | ActivityNet-TimeLens | | | QVHighlights-TimeLens | | |
|---|---|---|---|---|---|---|---|---|---|
| | R@0.3 | R@0.5 | R@0.7 | R@0.3 | R@0.5 | R@0.7 | R@0.3 | R@0.5 | R@0.7 |
| *Proprietary Models* | | | | | | | | | |
| GPT-4o (Hurst et al., 2024) | 60.6 | 44.5 | 23.5 | 55.2 | 41.4 | 25.8 | 69.0 | 54.8 | 38.5 |
| GPT-5 (Singh et al., 2025) | 59.3 | 42.0 | 22.0 | 57.4 | 44.9 | 30.4 | 72.4 | 60.4 | 46.4 |
| Gemini-2.0-Flash (Comanici et al., 2025) | 66.4 | 53.5 | 27.1 | 62.9 | 54.0 | 37.7 | 76.2 | 66.4 | 48.3 |
| Gemini-2.5-Flash (Comanici et al., 2025) | 68.7 | 56.1 | 30.6 | 66.8 | 57.5 | 41.3 | 78.2 | 69.4 | 55.0 |
| Gemini-2.5-Pro (Comanici et al., 2025) | 74.1 | 61.1 | 34.0 | 72.3 | 64.2 | 47.1 | 84.1 | 75.9 | 61.1 |
| *Open-Source Models* | | | | | | | | | |
| VideoChat-Flash-7B (Li et al., 2025d) | 60.2 | 37.9 | 17.8 | 35.5 | 21.8 | 10.5 | 45.2 | 30.6 | 16.7 |
| Qwen2.5-VL-7B$^\star$ (Bai et al., 2025b) | 58.1 | 35.1 | 18.2 | 47.2 | 32.5 | 20.2 | 55.0 | 41.7 | 29.3 |
| VideoChat-R1-7B (Li et al., 2025e) | 51.9 | 30.8 | 11.7 | 35.0 | 23.9 | 11.3 | 29.3 | 19.1 | 9.4 |
| Time-R1-7B (Wang et al., 2025) | 57.9 | 32.0 | 16.9 | 44.8 | 31.0 | 19.0 | 65.8 | 51.5 | 36.1 |
| TVG-R1-7B$^\star$ (Chen et al., 2025) | 44.5 | 23.6 | 12.4 | 46.7 | 31.0 | 18.6 | 55.8 | 41.2 | 28.0 |
| VideoChat-R1.5-7B$^\star$ (Yan et al., 2025) | 46.4 | 24.0 | 10.4 | 40.6 | 25.3 | 16.4 | 62.2 | 44.5 | 28.3 |
| MiMo-VL-7B (Xiaomi & Team, 2025) | 57.9 | 42.6 | 20.5 | 49.3 | 38.7 | 22.4 | 57.1 | 42.6 | 28.4 |
| Qwen3-VL-8B-Instruct$^\star$ (Bai et al., 2025a) | 61.7 | 41.5 | 23.1 | 41.2 | 30.7 | 20.0 | 46.6 | 38.2 | 29.5 |
| *Post-Training Frameworks* | | | | | | | | | |
| OP-RKD [Qwen3-VL-8B] | 67.3 | 47.0 | 27.8 | 59.1 | 44.9 | 30.6 | 70.7 | 57.4 | 44.5 |
| OP-FKD [Qwen3-VL-8B] | 66.9 | 47.3 | 27.6 | 58.6 | 44.2 | 30.5 | 71.2 | 58.9 | 46.5 |
| GRPO [Qwen3-VL-8B] | 72.7 | 44.4 | 27.6 | 58.6 | 42.7 | 32.1 | 69.8 | 53.0 | 41.5 |
| Video-OPD [Qwen3-VL-8B] (Ours) | **73.1** | **45.8** | **32.4** | **60.5** | **45.6** | **35.8** | **73.8** | **60.3** | **50.4** |

mentary components: Teacher Reliability Pre-Validation (TRPV) and Disagreement-Based Trajectory Prioritization (DBTP). Together, these mechanisms guide the selection and prioritization of informative on-policy trajectories, thereby improving sample efficiency and accelerating convergence. We describe each component in detail below.

**Step 1: Teacher Reliability Pre-Validation.** TRPV is designed to mitigate the impact of erroneous teacher signals in long-horizon temporal video grounding. Instead of directly supervising the student, ground-truth temporal annotations are used solely as a validation oracle to assess the temporal reliability of the teacher for each video–query pair. Concretely, a sample is deemed teacher-reliable if the temporal boundary predicted by the teacher satisfies a predefined consistency criterion with respect to the annotated ground-truth boundary. Samples that fail this validation are excluded from subsequent curriculum prioritization, thereby preventing unreliable teacher feedback from propagating into on-policy optimization. In practice, the consistency criterion is quantified by the mean IoU between teacher's top-$k$ predicted temporal boundaries and ground-truth annotation.

**Step 2: Disagreement-Based Trajectory Prioritization.** Conditioned on validated reliable samples, DBTP prioritizes trajectories that are most informative for the current student policy. Informativeness is characterized by the degree of teacher–student disagreement, quantified as the aggregated reverse KL divergence between the student policy and the fixed teacher along the sampled reasoning trajectory. Larger divergence values indicate a greater mismatch between the

student's current behavior and a reliable teacher signal, and therefore a higher potential for effective knowledge transfer through on-policy distillation.

TVDF is applied iteratively throughout training and naturally adapts as the student policy evolves. As optimization progresses, the distribution of high-disagreement trajectories shifts accordingly, allowing the curriculum to continuously target samples that remain both challenging for the student and reliable for the teacher. By leveraging temporal annotations solely as a validation signal, TVDF provides a principled and indirect means of supervision, improving sample efficiency and accelerating convergence. For additional implementation details, please refer to Section B.

## 4. Experiment

Section 4.1 describes the experimental setup. Section 4.2 presents the performance of Video-OPD on both TVG and broader video understanding tasks. Section 4.3 provides ablation studies that analyze the contributions of proposed Video-OPD framework and TVDF training curriculum. Additional experimental results are provided in Section D.

### 4.1. Experimental Setup

**Training Details.** We adopt Qwen3-VL-8B-Instruct (Bai et al., 2025a) as the base model. The maximum input video token length is set to 8,192, with videos sampled at 2 FPS. The number of video frames is capped at 768, and the maximum video frame token length is set to 768. During Video-

OPD training, we use a learning rate of $1 \times 10^{-6}$, a batch size of 32, and train for a single epoch. Qwen3-VL-32B post-trained with GRPO is employed as the teacher model. Implementation details are provided in Section D.1. For GRPO training, learning rate is set to $1 \times 10^{-6}$, batch size is 32, and 8 rollouts are performed per training instance.

**Construction of the Training Dataset.** Training video data are collected from multiple public datasets, including HiREST (Zala et al., 2023), QuerYD (Oncescu et al., 2021), HowTo-Interlink7M (Jinpeng Wang et al., 2024), VTimeLLM (Huang et al., 2024), and DiDeMo (Anne Hendricks et al., 2017). In addition, we incorporate temporal annotations from TimeLens-100K (Zhang et al., 2025), yielding a total of 96,586 temporal video grounding instances. For Video-OPD training, we apply TVDF as the data filtering strategy and select 2,500 samples to enable efficient optimization. For GRPO training, we follow the difficulty-aware data selection strategy proposed in Time-R1 (Wang et al., 2025) and similarly select 2,500 samples.

**Evaluation Benchmarks and Metrics.** We comprehensively evaluate our model on a diverse set of benchmarks spanning both TVG and general video understanding tasks. For TVG, we evaluate on Charades-STA (Gao et al., 2017), ActivityNet (Caba Heilbron et al., 2015), and QVHighlights (Lei et al., 2021), using the corrected temporal annotations from TimeLens (Zhang et al., 2025) to address errors in the original evaluation sets, and report the IoU between predicted temporal boundaries and ground-truth annotations, including Recall at IoU thresholds of $0.3, 0.5, 0.7$ and mean IoU, following prior work (Li et al., 2025a; Wang et al., 2024a;b). For general video understanding, we evaluate on TempCompass (Liu et al., 2024), MVBench (Li et al., 2024), and Video-MME (Fu et al., 2025), using accuracy as the evaluation metric across all benchmarks.

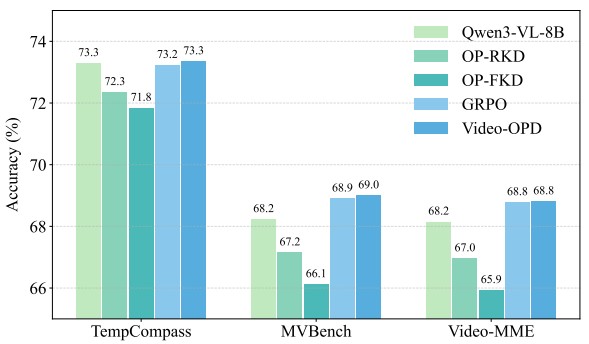

*Figure 4.* Video-OPD on broader video understanding tasks.

### 4.2. Main Results

**Temporal Video Grounding.** As illustrated in Table 1, Video-OPD achieves state-of-the-art (SOTA) performance across all benchmarks among open-source models. Notably,

Video-OPD consistently approaches the performance of the closed-source model Gemini-2.5-Flash on most datasets, and even surpasses it on several benchmarks. Moreover, when compared with existing post-training frameworks, including GRPO, Off-Policy Forward KL Distillation (OP-FKD), and Off-Policy Reverse KL Distillation (OP-RKD), Video-OPD demonstrates substantial and consistent performance gains. Implementation details of the off-policy distillation methods are provided in Section C.

*Table 2.* Ablation study of proposed TVDF training curriculum. $^{\star}$ indicates that data have been re-annotated using TimeLens.

| TVDF | | Charades$^{\star}$ | | ActivityNet$^{\star}$ | | QVHighlights$^{\star}$ | |
|---|---|---|---|---|---|---|---|
| TRPV | DBTP | R@0.7 | mIoU | R@0.7 | mIoU | R@0.7 | mIoU |
| Base Model | | 23.1 | 42.9 | 20.0 | 30.4 | 29.5 | 36.9 |
| ✗ | ✗ | 31.1 | 51.2 | 33.5 | 45.7 | 47.7 | 58.7 |
| ✓ | ✗ | 31.7 | 51.7 | 34.2 | 46.4 | 48.2 | 59.5 |
| ✗ | ✓ | 31.3 | 51.4 | 34.6 | 46.9 | 50.0 | 60.3 |
| ✓ | ✓ | 32.4 | 52.0 | 35.8 | 47.3 | 50.4 | 61.0 |

**General Video Understanding.** As illustrated in Figure 4, off-policy distillation leads to noticeable performance degradation on a broader range of video understanding tasks. In contrast, both GRPO and Video-OPD not only preserve performance but also yield modest performance improvements. Notably, Video-OPD consistently achieves SOTA results across all evaluated datasets, demonstrating strong generalization capability beyond TVG.

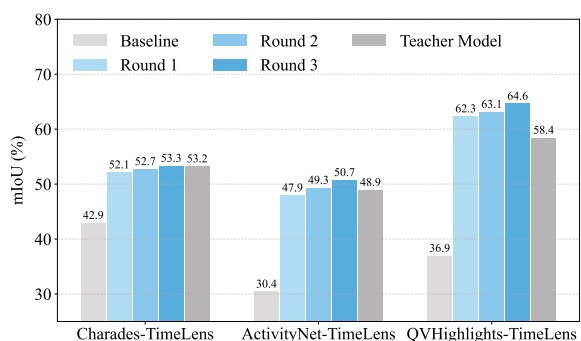

*Figure 5.* Performance of Video-OPD under multi-round training.

### 4.3. Ablation Study

**Effectiveness of the TVDF Training Curriculum.** We conduct ablation studies on three TVG benchmarks to systematically assess the effectiveness of proposed TVDF training curriculum. As shown in Table 2, incorporating TVDF yields an additional performance gain of approximately 2% for Video-OPD, demonstrating its ability to precisely identify and prioritize highly effective training samples. Moreover, both TRPV and DBTP contribute positively to the overall performance, confirming that each component is individually effective and that they complement each other

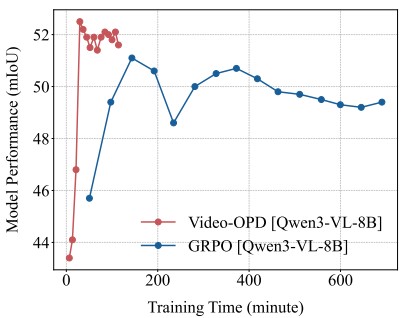 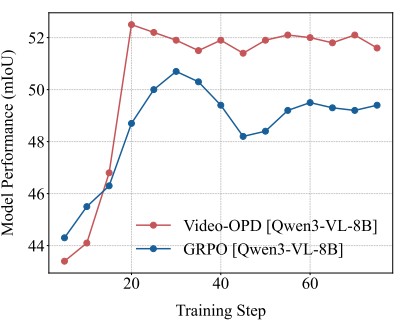 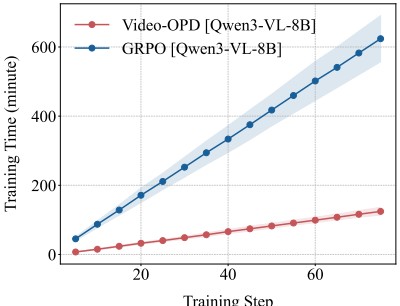

*Figure 6.* Training convergence behavior and computational cost of Video-OPD and GRPO, evaluated on Charades-TimeLens benchmark.

in improving optimization efficiency.

**Video-OPD under Multi-Round Training.** As discussed in Section 3.2, TVDF training curriculum can be applied iteratively throughout training. As illustrated in Figure 5, Video-OPD achieves consistent performance gains as the number of training rounds increases. This behavior indicates that TVDF effectively adapts to the evolving student policy, continuously prioritizing trajectories that remain both challenging for the student and reliable for the teacher. Notably, after three training rounds, Video-OPD consistently surpasses the teacher model, demonstrating that its performance is not intrinsically constrained by that of the teacher. For the complete ablation results of multi-round Video-OPD training, please refer to Table 6.

*Table 3.* Performance of Video-OPD under different teacher models. Models marked with $^\star$ are post-trained using GRPO. Experiments are conducted on QVHighlights-TimeLens dataset.

| Models for Evaluation | R@0.3 | R@0.5 | R@0.7 | mIoU |
|---|---|---|---|---|
| Qwen3-VL-8B-Instruct | 46.6 | 38.2 | 29.5 | 36.9 |
| Teacher: Qwen3-VL-4B $^\star$ | 70.5 | 53.3 | 41.8 | 54.9 |
| Student: Qwen3-VL-8B | **75.6** | **62.8** | **51.9** | **62.5** |
| *Performance Gap* | *+5.1* | *+9.5* | *+10.1* | *+7.6* |
| Teacher: Qwen3-VL-8B $^\star$ | 69.8 | 53.0 | 41.5 | 54.3 |
| Student: Qwen3-VL-8B | **74.9** | **62.0** | **51.8** | **62.3** |
| *Performance Gap* | *+5.1* | *+9.0* | *+10.3* | *+8.0* |
| Teacher: Qwen3-VL-32B $^\star$ | 77.1 | 62.8 | 50.9 | 62.8 |
| Student: Qwen3-VL-8B | **75.9** | **63.6** | **52.0** | **62.9** |
| *Performance Gap* | *-1.2* | *+0.8* | *+1.1* | *+0.1* |

**Performance of Video-OPD under Different Teacher.** As shown in Table 3, stronger teacher models consistently lead to larger performance gains for Video-OPD, indicating that the student benefits increasingly from higher-quality teacher supervision. This trend demonstrates that Video-OPD can effectively scale with improvements in teacher capability, exhibiting a sustainable and forward-compatible learning behavior. Notably, in nearly all settings, the final student model matches or even surpasses the corresponding teacher, confirming that the performance of Video-OPD is not inher-

ently bounded by the teacher's capacity. For the complete ablation results of the teacher model, please refer to Table 5.

## 5. Extended Analysis

In Section 2.2, we identify two fundamental limitations of GRPO: inefficient optimization arising from sparse reward signals and substantial computational overhead due to multiple rollouts. In this section, we systematically compare the training dynamics of Video-OPD and GRPO, demonstrating that the performance gains of Video-OPD stem entirely from its effective mitigation of these two limitations.

**Dense Supervision Improves Optimization Efficiency.** As shown in the left and middle panels of Figure 6, Video-OPD converges to an optimal policy substantially faster than GRPO during training, while also achieving superior overall performance. These results confirm that Video-OPD provides precise, dense, and fine-grained learning signals, enabling more accurate credit assignment and thereby improving both optimization efficiency and effectiveness.

**Eliminating Multiple Rollouts Reduces Computational Overhead.** As shown in the right panel of Figure 6, under the same number of training steps, Video-OPD requires significantly less time than GRPO, approximately only 20% of its training cost. This result confirms that, by avoiding multi-rollout reward estimation for variance reduction, Video-OPD dramatically lowers computational overhead while maintaining stable optimization.

## 6. Related Work

**MLLMs for Temporal Video Grounding (TVG).** A substantial body of prior work has investigated methods for enhancing the TVG capabilities of multimodal large language models (MLLMs). One research direction focuses on training paradigms, including augmenting supervised fine-tuning with TVG-specific objectives (Li et al., 2025a; Zeng et al., 2024) and designing verifiable reward signals to enable reinforcement learning–based optimization (Chen et al., 2025; Li et al., 2025f; Wang et al., 2025; Yue et al.,

2025; Zhang et al., 2025). A complementary line of work explores architectural innovations, such as token compression mechanisms to alleviate the computational burden of long video sequences (Ren et al., 2024) and timestamp encoding schemes that explicitly align frame-level representations with their temporal positions (Chen et al., 2024; Ge et al., 2025; Wu et al., 2025; Zeng et al., 2025; Li et al., 2025g).

**On-Policy Distillation.** On-policy distillation (Lu & Lab, 2025) is a recent post-training paradigm that unifies the advantages of on-policy reinforcement learning and knowledge distillation by jointly enforcing training–inference distributional alignment and dense token-level supervision. Recent large-scale models, including MiMo-V2-Flash (Xiao et al., 2026), Qwen3 (Yang et al., 2025), and Qwen3-VL (Bai et al., 2025a), adopt on-policy distillation to align a student model with multiple RL-enhanced, domain-specialized teachers through token-level reverse KL divergence computed over trajectories sampled from the student's current policy. This design enables efficient transfer of multi-domain expertise while mitigating off-policy distributional mismatch. To our knowledge, this work makes the first attempt to introduce on-policy distillation to TVG, addressing the challenges of long-horizon temporal reasoning that require both strict distributional alignment and fine-grained credit assignment.

## 7. Conclusion

We introduce Video-OPD, a novel post-training framework for TVG that performs on-policy optimization over trajectories sampled from the current policy and provides dense, token-level supervision in place of sparse, trajectory-level rewards. Building upon this framework, we propose Teacher-Validated Disagreement Focusing (TVDF), a lightweight curriculum strategy that further improves optimization efficiency by leveraging annotated temporal data solely as a validation signal to prioritize trajectories that are both teacher-reliable and maximally informative. Experiments demonstrate that Video-OPD consistently outperforms GRPO, achieving faster convergence and lower computational cost.

## Limitations

Video-OPD assumes access to a high-capacity teacher model capable of providing conditional log-probabilities for student-generated trajectories, which introduces practical considerations regarding teacher availability and reliability. Importantly, however, Video-OPD does not rely on an extremely large, general-purpose language model with broad capabilities. In practice, it can instead perform online distillation from multiple domain-specific expert models, enabling comprehensive improvements in video understanding across targeted tasks. In our experiments, the GRPO-trained Qwen3-VL-4B-Instruct serves as a lightweight expert spe-

cialized for temporal video grounding. Moreover, after multiple rounds of Video-OPD training, the student model consistently surpasses its teacher. Taken together, these observations demonstrate that Video-OPD is both practically feasible and broadly applicable.

## Impact Statement

This work advances temporal video grounding by improving the efficiency, stability, and scalability of post-training frameworks through on-policy distillation. By eliminating sparse rewards and computationally expensive rollouts, Video-OPD significantly reduces the cost of training multimodal large language models, thereby lowering barriers for academic researchers and resource-constrained practitioners. Progress in TVG can benefit downstream applications such as video retrieval, human–computer interaction, and embodied AI. As with all video–language systems, responsible deployment is essential to mitigate risks related to biased data, privacy concerns, and potential misuse in surveillance or content manipulation.

## Acknowledgement.

This work makes use of the TimeLens-Bench and TimeLens-100K datasets (https://github.com/TencentARC/TimeLens ), which are licensed under the License Term of TimeLens. The authors of this work confirm that the use of the above datasets in this work is strictly limited to academic research purposes and does not involve any commercial activities.

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

## A. Proof Sketch: Dense Rewards and Optimization Advantages of Video-OPD over GRPO

We sketch why Video-OPD induces strictly denser rewards than GRPO and why such density leads to faster and more stable optimization.

**Reward Density.** In GRPO, each trajectory $\tau = (a_1, \ldots, a_T)$ is associated with a single sequence-level reward $R(\tau)$, typically derived from a timestamp-aware IoU. Ignoring the KL regularization term for clarity, the corresponding policy gradient can be written as

$$\nabla_\theta J_{\mathrm{GRPO}} = \mathbb{E}_{\tau \sim \pi_{\theta_{\mathrm{old}}}} \Big[ R(\tau) \sum_{t=1}^{T} \nabla_\theta \log \pi_\theta(a_t \mid s_t) \Big]. \tag{7}$$

Since $R(\tau)$ is independent of $t$, the same scalar reward is uniformly assigned to all time steps, yielding a temporally sparse and degenerate supervision signal that provides no explicit guidance for individual intermediate decisions. In contrast, Video-OPD defines a token-level reward at each time step

$$r_t = -\big( \log \pi_\theta(a_t \mid s_t) - \log \pi_{\mathrm{tea}}(a_t \mid s_t) \big), \tag{8}$$

evaluated on trajectories sampled from the current student policy. The resulting policy gradient estimator is

$$\nabla_\theta J_{\mathrm{OPD}} = \mathbb{E}_{\tau \sim \pi_{\theta_{\mathrm{old}}}} \Big[ \sum_{t=1}^{T} r_t \, \nabla_\theta \log \pi_\theta(a_t \mid s_t) \Big]. \tag{9}$$

Therefore, Video-OPD provides $T$ distinct, step-specific rewards per trajectory, strictly increasing reward density along the temporal dimension.

**Variance Reduction and Credit Assignment.** The variance of a policy gradient estimator admits the decomposition

$$\mathrm{Var}\bigg( \sum_{t=1}^{T} A_t \nabla_\theta \log \pi_t \bigg) = \sum_{t=1}^{T} \mathrm{Var}(A_t \nabla_\theta \log \pi_t) + 2 \sum_{t < t'} \mathrm{Cov}(A_t \nabla_\theta \log \pi_t, \, A_{t'} \nabla_\theta \log \pi_{t'}). \tag{10}$$

For GRPO, $A_t \equiv R(\tau)$ for all $t$, which induces strong positive correlations across time steps. As a result, the gradient variance grows rapidly with the trajectory length $T$, and sequence-level rewards offer no mechanism for precise temporal credit assignment.

For Video-OPD, each $A_t = r_t$ depends only on the local state–action pair $(s_t, a_t)$. Since the teacher policy $\pi_{\mathrm{tea}}$ is fixed and evaluated on the student's state distribution, the resulting rewards are well-conditioned and largely decorrelated across time. Consequently, the cross-time covariance terms are substantially reduced, yielding a lower-variance gradient estimator.

**Relation to KL Minimization.** Moreover, the expected Video-OPD update is equivalent to minimizing the expected reverse KL divergence between the student and teacher policies:

$$\mathbb{E}_{a_t \sim \pi_\theta} \big[ r_t \nabla_\theta \log \pi_\theta(a_t \mid s_t) \big] = \nabla_\theta D_{\mathrm{KL}} \big( \pi_\theta(\cdot \mid s_t) \,\|\, \pi_{\mathrm{tea}}(\cdot \mid s_t) \big). \tag{11}$$

This shows that Video-OPD performs on-policy stochastic gradient descent on a smooth objective at each visited state, further contributing to stable optimization.

**Implications for Convergence.** For smooth objectives optimized via stochastic gradients, the expected optimality gap decreases at a rate inversely proportional to the gradient variance. Since Video-OPD yields strictly lower-variance gradient estimates than GRPO while preserving strict on-policy optimization, it converges faster under the same sampling budget. In addition, the dense token-level rewards enable accurate credit assignment, leading to improved final performance.

$$\square$$

## B. Additional Implementation Details of the Teacher-Validated Disagreement Focusing.

In practice, teacher reliability is measured by the mean IoU between the teacher's top-$k$ predicted temporal intervals and the annotated ground-truth interval. Meanwhile, we compute the corresponding mean IoU for the student model and use their discrepancy to guide data selection. This design emphasizes samples where the teacher prediction is reliable while the student exhibits noticeable disagreement, thereby improving the effectiveness of knowledge transfer.

Formally, let $\mathcal{D} = \{(x_i, y_i)\}_{i=1}^N$ denote the training dataset. For each sample $x_i$, we compute the teacher IoU $\tau_i$ and the student IoU $\sigma_i$, and define the teacher–student IoU difference as

$$\delta_i = \tau_i - \sigma_i. \tag{12}$$

Let $\mathcal{S} = \{s_1, s_2, \ldots, s_n\}$ denote the set of matched samples associated with $\delta_i$. Our objective is to select a subset of $k$ samples ($k \leq n$) from $\mathcal{S}$ for training. We consider the following difference-based sampling strategies.

### B.1. Difference-Sorted Uniform Sampling (DSUS)

This strategy samples uniformly across the entire spectrum of IoU differences to preserve diversity. Samples are first sorted in descending order of $\delta_i$:

$$\tilde{\mathcal{S}} = (s_{(1)}, s_{(2)}, \ldots, s_{(n)}), \quad \text{where } \delta_{(1)} \geq \delta_{(2)} \geq \cdots \geq \delta_{(n)}. \tag{13}$$

We then select $k$ evenly spaced indices

$$j_t = \left\lfloor 1 + (n-1)\frac{t}{k-1} \right\rfloor, \quad t = 0, 1, \ldots, k-1, \tag{14}$$

and construct the sampled set as

$$\mathcal{S}_{\text{DSUS}} = \{s_{(j_t)} \mid t = 0, \ldots, k-1\}. \tag{15}$$

### B.2. Top-$k$ Difference-Based Sampling (Top-$k$ DBS)

This strategy directly selects samples with the largest teacher–student disagreement. Using the same sorted sequence $\tilde{\mathcal{S}}$, the sampled subset is defined as

$$\mathcal{S}_{\text{Top-k}} = \{s_{(1)}, s_{(2)}, \ldots, s_{(k)}\}. \tag{16}$$

This selection focuses training on samples with the greatest IoU discrepancies, aligning with our objective of emphasizing the most informative high-disagreement cases. Unless otherwise specified, this strategy is adopted throughout all experiments.

### B.3. Bucket-Balanced Difference Sampling (BBDS)

To promote balanced coverage across different disagreement levels, we partition the IoU difference range into $B$ disjoint buckets. Let

$$\delta_{\min} = \min_i \delta_i, \qquad \delta_{\max} = \max_i \delta_i. \tag{17}$$

The $b$-th bucket is defined as

$$\mathcal{B}_b = \left\{ s_i \in \mathcal{S} \ \middle| \ \delta_{\min} + (b-1)\frac{\delta_{\max} - \delta_{\min}}{B} \leq \delta_i < \delta_{\min} + b\frac{\delta_{\max} - \delta_{\min}}{B} \right\}, \quad b = 1, \ldots, B, \tag{18}$$

with the last bucket including $\delta_{\max}$.

Given a total sampling budget $k$, each bucket is allocated

$$k_b = \left\lfloor \frac{k}{B} \right\rfloor + \begin{cases} 1, & b \leq (k \bmod B), \\ 0, & \text{otherwise,} \end{cases} \tag{19}$$

samples. Within each non-empty bucket $\mathcal{B}_b$, samples are sorted in descending order of $\delta_i$, and $k_b$ samples are selected using uniform spacing as in Difference-Sorted Uniform Sampling. The final sampled set is obtained by

$$\mathcal{S}_{\text{BBDS}} = \bigcup_{b=1}^{B} \mathcal{S}_b. \tag{20}$$

In our experiments, we set $B = 5$.

### B.4. Gaussian-Weighted Difference Sampling (GWDS)

We consider a probabilistic sampling strategy based on a Gaussian weighting over IoU differences. Given a target difference center $c$ and standard deviation $\sigma$, the sampling probability for each sample is defined as

$$p_i = \frac{1}{Z} \exp\left( -\frac{(\delta_i - c)^2}{2\sigma^2} \right), \quad Z = \sum_{j=1}^{n} \exp\left( -\frac{(\delta_j - c)^2}{2\sigma^2} \right). \tag{21}$$

A subset of $k$ samples is then drawn without replacement from the categorical distribution specified by $\{p_i\}_{i=1}^{n}$:

$$\mathcal{S}_{\text{GWDS}} \sim \text{Categorical}(\{p_i\}, k). \tag{22}$$

In our experiments, we use $c = 0.9$ and $\sigma = 0.2$.

## C. Off-Policy Distillation Frameworks for Temporal Video Grounding

We examine two purely off-policy distillation regimes that operate on fixed supervised trajectories, without any on-policy sampling. Both regimes follow the same training pipeline as Video-OPD, differing only in the source of trajectories: ground-truth sequences from the training corpus are used in place of student-generated rollouts. Specifically, we consider two frameworks: Off-Policy Reverse KL Distillation (OP-RKD) and Off-Policy Forward KL Distillation (OP-FKD).

### C.1. Off-Policy Reverse KL Distillation (OP-RKD)

We first consider an off-policy distillation regime that adopts reverse KL divergence as the supervision signal, using the same divergence choice as Video-OPD but operating entirely on fixed ground-truth trajectories.

**Step 1: Fixed Trajectory Selection.** Given a video-query pair $(v, q)$, we select a target sequence $\tau = (a_1, \ldots, a_T)$ from the training corpus. Unlike Video-OPD, which samples trajectories from the current student policy $\pi_\theta$, this regime uses ground-truth sequences that are independent of the student's current parameters. During this selection, we record the student's token-level log-probabilities $\log \pi_\theta(a_t \mid s_t)$ for each token in the ground-truth sequence, where $s_t = (v, q, a_{<t})$, which will later be used for computing per-token optimization signals.

**Step 2: Evaluation on Fixed Trajectories.** For the same trajectory $\tau$, we query a fixed, high-capacity teacher model $\pi_{\text{tea}}$ to compute the conditional log-probabilities

$$\log \pi_{\text{tea}}(a_t \mid a_{<t}, v, q), \tag{23}$$

for each token in the ground-truth sequence. Importantly, the teacher never generates tokens itself and is only used to evaluate the fixed ground-truth sequences. Consequently, the supervision provided by the teacher is aligned with the corpus distribution rather than the student's on-policy state distribution, making this regime vulnerable to distributional mismatch between training and inference.

**Step 3: Dense Token-Level Supervision.** Using the student and teacher log-probabilities, we define a dense per-token signal induced by the reverse KL divergence:

$$r_t = -\Big( \log \pi_\theta(a_t \mid s_t) - \log \pi_{\text{tea}}(a_t \mid s_t) \Big), \tag{24}$$

which is the pointwise contribution encouraging $\pi_\theta$ to match $\pi_{\text{tea}}$ at state $s_t = (v, q, a_{<t})$. This is identical in form to Video-OPD's Step 3, but computed on ground-truth tokens $a_t$ rather than student-sampled tokens $a_t$. Tokens assigned low probability by the teacher incur larger penalties under this signal, yielding fine-grained supervision that precisely identifies intermediate reasoning steps deviating from the teacher's preferred behavior. By providing supervision at every generation step, Off-Policy Reverse-KL Distillation converts sparse, delayed episode-level rewards into dense learning signals, enabling precise temporal credit assignment across long reasoning chains, just as in Video-OPD.

**Step 4: Policy Update with Dense Advantages.** The student policy is updated via standard policy-gradient optimization, where the teacher-derived signal $r_t$ is treated as a token-level advantage. Concretely, the policy parameters $\theta$ are updated according to

$$\nabla_\theta \mathcal{L}_{\text{OP-RKD}} = -\mathbb{E}_{\tau \sim \mathcal{D}} \left[ \sum_{t=1}^{T} r_t \frac{\pi_\theta(a_t \mid s_t)}{\pi_{\theta_{\text{old}}}(a_t \mid s_t)} \nabla_\theta \log \pi_\theta(a_t \mid s_t) \right], \tag{25}$$

where $\mathcal{D}$ denotes the distribution over fixed ground-truth trajectories from the training corpus, and each decision $y_t$ is directly supervised by a time-dependent advantage $r_t$. In contrast to Video-OPD, which samples trajectories from the current student policy $\pi_\theta$, this regime uses trajectories from the fixed corpus distribution $\mathcal{D}$, making the optimization strictly off-policy.

## C.2. Off-Policy Forward KL Distillation (OP-FKD)

We further consider a vanilla knowledge distillation regime that adopts the classical forward KL divergence. The training pipeline follows the same four-stage structure as OP-RKD.

**Step 1: Fixed Trajectory Selection.** As in OP-RKD, given a video-query pair $(v, q)$, we select a target sequence $\tau = (a_1, \ldots, a_T)$ from the training corpus. Unlike Video-OPD, which samples trajectories from the current student policy $\pi_\theta$, this regime uses ground-truth sequences that are independent of the student's current parameters. During this selection, we record the student's token-level probability distribution $P_{\text{stu}}(\cdot \mid s_t)$ for each state $s_t = (v, q, a_{<t})$ in the ground-truth sequence, which will later be used for computing per-token optimization signals.

**Step 2: Evaluation on Fixed Trajectories.** For the same trajectory $\tau$, we query a fixed, high-capacity teacher model $\pi_{\text{tea}}$ to compute the conditional probability distribution

$$P_{\text{tea}}(\cdot \mid s_t) = \pi_{\text{tea}}(\cdot \mid a_{<t}, v, q), \tag{26}$$

for each state $s_t$ in the ground-truth sequence. Importantly, the teacher never generates tokens itself and is only used to evaluate the fixed ground-truth sequences.

Consequently, the supervision provided by the teacher is aligned with the corpus distribution rather than the student's on-policy state distribution, making this regime vulnerable to distributional mismatch between training and inference.

**Step 3: Dense Token-Level Supervision.** Using the student and teacher probability distributions, we define a dense per-token signal induced by the forward KL divergence:

$$\ell_t = \text{KL}\big(P_{\text{tea}}(\cdot \mid s_t) \,\big\|\, P_{\text{stu}}(\cdot \mid s_t)\big) = \sum_{w \in \mathcal{V}} P_{\text{tea}}(w \mid s_t) \log \frac{P_{\text{tea}}(w \mid s_t)}{P_{\text{stu}}(w \mid s_t)}, \tag{27}$$

where $\mathcal{V}$ denotes the vocabulary, $P_{\text{tea}}(\cdot \mid s_t)$ and $P_{\text{stu}}(\cdot \mid s_t)$ are the teacher and student token distributions at state $s_t = (v, q, a_{<t})$, and $w$ represents a token in the vocabulary. Forward KL is inherently mode-covering: it encourages the student to allocate probability mass wherever the teacher assigns non-negligible probability, including low-probability modes, thereby promoting a more conservative and diversified approximation of the teacher. In contrast to reverse KL used in Video-OPD and OP-RKD, forward KL reverses the order of the distributions in the divergence, leading to different optimization dynamics that emphasize broader coverage of the teacher's distribution rather than concentration on its peaks.

**Step 4: Standard Supervised Learning Update.** The student policy is updated via standard supervised learning, where the forward KL divergence serves as the per-token loss. Since the teacher distribution $P_{\text{tea}}$ is fixed with respect to the student parameters $\theta$, the gradient of the loss with respect to $\theta$ simplifies to:

$$\nabla_\theta \ell_t = -\sum_{w \in \mathcal{V}} P_{\text{tea}}(w \mid s_t) \nabla_\theta \log P_{\text{stu}}(w \mid s_t). \tag{28}$$

Concretely, the policy parameters $\theta$ are updated according to

$$\nabla_\theta \mathcal{L}_{\text{OP-FKD}} = \mathbb{E}_{\tau \sim \mathcal{D}} \left[ \sum_{t=1}^{T} \nabla_\theta \ell_t \right], \tag{29}$$

where $\mathcal{D}$ denotes the distribution over fixed ground-truth trajectories from the training corpus. In contrast to Video-OPD and OP-RKD, which use policy-gradient optimization with token-level advantages, this regime uses standard backpropagation through the divergence loss.

# D. More Details Regarding the Experiments on Video-OPD and TVDF

In Section D.1, we provide a detailed description of the practical implementation of GRPO training. In Section D.2, we compare the performance of Video-OPD and TimeLens on the TVG task. Sections D.2 to D.6 present comprehensive ablation studies of Video-OPD and the proposed VTDF framework. Finally, Section D.7 illustrates the prompt templates used by Video-OPD during both training and inference.

## D.1. Detailed GRPO Implementation Details

Following prior work (Wang et al., 2025; Zhang et al., 2025), Qwen3-VL-4B-Instruct, Qwen3-VL-8B-Instruct, and Qwen3-VL-32B-Instruct are further trained using reinforcement learning with verifiable rewards (e.g., GRPO), yielding Qwen3-VL-4B-GRPO, Qwen3-VL-8B-GRPO, and Qwen3-VL-32B-GRPO, respectively. Unless otherwise specified, we adopt Qwen3-VL-32B-GRPO as the teacher model.

**Training Details.** For both training and evaluation under GRPO, the maximum input video token length is fixed at 8,192. Videos are sampled at 2 FPS, and both the maximum number of frames and the maximum frame token length are limited to 768. We use a learning rate of $1 \times 10^{-6}$, a total batch size of 32, and 8 rollouts.

**Dataset Construction.** We adopt the difficulty-aware data selection strategy proposed in Time-R1 and TimeLens to construct a training set of 2,500 samples. Specifically, we employ a Gaussian-based sampling scheme over the Intersection-over-Union (IoU) scores of the data points. Each sample is selected according to a probability defined by a Gaussian distribution centered at a specified mean $\mu$, computed as::

$$P(i) = \frac{1}{Z} \exp\left(-\frac{(x_i - \mu)^2}{2\sigma^2}\right), \tag{30}$$

where $x_i$ represents the IoU value of the $i$-th data point, $\mu$ is the center of the Gaussian distribution, and $\sigma$ is the standard deviation. The normalization factor $Z$ is the sum of the unnormalized probabilities over all $n$ data points:

$$Z = \sum_{j=1}^{n} \exp\left(-\frac{(x_j - \mu)^2}{2\sigma^2}\right), \tag{31}$$

where $n$ denotes the total number of data points. We set the Gaussian mean and standard deviation to $\mu = 0.3$ and $\sigma = 0.2$.

*Table 4.* Comparison with TimeLens-8B. TimeLens-8B is supervised-finetuned before GRPO. **Bold** text indicates the best performance.

| Models for Evaluation | Charades-TimeLens | | | | ActivityNet-TimeLens | | | | QVHighlights-TimeLens | | | |
|---|---|---|---|---|---|---|---|---|---|---|---|---|
| | R@0.3 | R@0.5 | R@0.7 | mIoU | R@0.3 | R@0.5 | R@0.7 | mIoU | R@0.3 | R@0.5 | R@0.7 | mIoU |
| Qwen3-VL-8B-Instruct | 61.7 | 41.5 | 23.1 | 42.9 | 41.2 | 30.7 | 20.0 | 30.4 | 46.6 | 38.2 | 29.5 | 36.9 |
| TimeLens-8B | **74.6** | **49.5** | 33.4 | 53.3 | 63.8 | 48.0 | 36.4 | 49.3 | **77.8** | 63.4 | 51.8 | 63.0 |
| Video-OPD (Round 1) | 73.1 | 45.8 | 32.4 | 52.0 | 60.5 | 45.6 | 35.8 | 47.3 | 73.8 | 60.3 | 50.4 | 61.0 |
| Video-OPD (Round 2) | 73.3 | 47.4 | 33.4 | 52.7 | 63.1 | 47.3 | 37.4 | 49.3 | 75.9 | 63.1 | 53.2 | 63.1 |
| Video-OPD (Round 3) | 74.2 | 48.6 | **33.8** | **53.4** | **64.4** | **49.2** | **39.0** | **50.7** | 77.6 | **65.3** | **55.3** | **64.6** |

## D.2. Video-OPD vs. TimeLens: Performance Comparison

As shown in Table 4, we compare our method with the GRPO-trained state-of-the-art TimeLens-8B, which uses a maximum input video token length of 8192. Notably, TimeLens-8B is first supervisedly fine-tuned on temporal video grounding data and subsequently optimized with GRPO using 12k training samples, whereas Video-OPD-8B is trained purely via on-policy distillation with only 2.5k samples. Despite this substantially reduced training budget, Video-OPD-8B achieves comparable performance across all TVG benchmarks. Moreover, after three OPD rounds (7.5k samples in total), Video-OPD-8B consistently surpasses TimeLens-8B on every benchmark; for instance, it attains an mIoU of 64.6 on QVHighlights-TimeLens, exceeding TimeLens-8B's 63.0 by 1.6 points.

## D.3. Complete Ablation Results of Video-OPD under Different Teacher Models

As shown in Table 5, we perform a single round of On-Policy Distillation (OPD) using Qwen3-VL-4B-GRPO, Qwen3-VL-8B-GRPO, and Qwen3-VL-32B-GRPO as teachers, with Qwen3-VL-8B-Instruct as the student. When Qwen3-VL-4B-GRPO or Qwen3-VL-8B-GRPO serves as the teacher, a single OPD round is sufficient for the 8B student to consistently surpass the teacher across all benchmarks, highlighting the efficiency and effectiveness of OPD.

Concretely, with Qwen3-VL-4B-GRPO as teacher, Video-OPD-8B (4B Teacher) achieves an mIoU of 62.5 on QVHighlights-TimeLens, exceeding the teacher's 54.9 by 7.6 points; with Qwen3-VL-8B-GRPO as the teacher, Video-OPD-8B (8B Teacher) attains an mIoU of 47.0 on ActivityNet-TimeLens, outperforming the teacher's 44.7 by 2.3 points. In contrast, when Qwen3-VL-32B-GRPO is used as the teacher, a single OPD round allows Video-OPD-8B (32B Teacher) to surpass the teacher on QVHighlights-TimeLens, while remaining slightly below on ActivityNet-TimeLens and Charades-TimeLens.

*Table 5.* Comprehensive evaluation of Video-OPD under different teacher models. **Bold** indicate the performance of the student model after Video-OPD training.

| Models for Evaluation | Charades-TimeLens | | | | ActivityNet-TimeLens | | | | QVHighlights-TimeLens | | | |
|---|---|---|---|---|---|---|---|---|---|---|---|---|
| | R@0.3 | R@0.5 | R@0.7 | mIoU | R@0.3 | R@0.5 | R@0.7 | mIoU | R@0.3 | R@0.5 | R@0.7 | mIoU |
| Qwen3-VL-8B-Instruct | 61.7 | 41.5 | 23.1 | 42.9 | 41.2 | 30.7 | 20.0 | 30.4 | 46.6 | 38.2 | 29.5 | 36.9 |
| Teacher: Qwen3-VL-4B-GRPO | 72.4 | 45.7 | 28.5 | 49.8 | 58.3 | 41.6 | 29.9 | 43.8 | 70.5 | 53.3 | 41.8 | 54.9 |
| **Student: Qwen3-VL-8B** | **73.6** | **47.2** | **33.0** | **52.4** | **61.9** | **46.5** | **36.7** | **48.4** | **75.6** | **62.8** | **51.9** | **62.5** |
| *Performance Gap* | *+1.2* | *+1.5* | *+4.5* | *+2.6* | *+3.6* | *+4.9* | *+6.8* | *+4.6* | *+5.1* | *+9.5* | *+10.1* | *+7.6* |
| Teacher: Qwen3-VL-8B-GRPO | 72.7 | 44.4 | 27.6 | 49.6 | 58.6 | 42.7 | 32.1 | 44.7 | 69.8 | 53.0 | 41.5 | 54.3 |
| **Student: Qwen3-VL-8B** | **72.0** | **46.6** | **33.0** | **51.6** | **61.2** | **44.8** | **35.0** | **47.0** | **74.9** | **62.0** | **51.8** | **62.3** |
| *Performance Gap* | *-0.7* | *+2.2* | *+5.4* | *+2.0* | *+2.6* | *+2.1* | *+2.9* | *+2.3* | *+5.1* | *+9.0* | *+10.3* | *+8.0* |
| Teacher: Qwen3-VL-32B-GRPO | 78.8 | 51.0 | 32.6 | 55.1 | 66.3 | 50.2 | 38.9 | 51.5 | 77.1 | 62.8 | 50.9 | 62.8 |
| **Student: Qwen3-VL-8B** | **73.3** | **48.4** | **33.6** | **52.5** | **62.2** | **45.6** | **35.8** | **47.7** | **75.9** | **63.6** | **52.0** | **62.9** |
| *Performance Gap* | *-5.5* | *-2.6* | *+1.0* | *-2.6* | *-4.1* | *-4.6* | *-3.1* | *-3.8* | *-1.2* | *+0.8* | *+1.1* | *+0.1* |

*Table 6.* Multi-round training performance of Video-OPD on three TVG benchmarks. **Bold** values indicate the best performance.

| Models for Evaluation | Charades-TimeLens | | | | ActivityNet-TimeLens | | | | QVHighlights-TimeLens | | | |
|---|---|---|---|---|---|---|---|---|---|---|---|---|
| | R@0.3 | R@0.5 | R@0.7 | mIoU | R@0.3 | R@0.5 | R@0.7 | mIoU | R@0.3 | R@0.5 | R@0.7 | mIoU |
| Qwen3-VL-32B-GRPO | **78.7** | **49.4** | 29.4 | 53.2 | **65.0** | 47.5 | 35.0 | 48.9 | 75.6 | 58.2 | 44.9 | 58.4 |
| Qwen3-VL-8B-Instruct | 61.7 | 41.5 | 23.1 | 42.9 | 41.2 | 30.7 | 20.0 | 30.4 | 46.6 | 38.2 | 29.5 | 36.9 |
| Video-OPD (round 1) | 73.0 | 45.9 | 32.5 | 52.1 | 61.2 | 46.0 | 36.1 | 47.9 | 75.2 | 62.3 | 51.9 | 62.3 |
| Video-OPD (round 2) | 73.3 | 47.4 | 33.4 | 52.7 | 63.1 | 47.3 | 37.4 | 49.3 | 75.9 | 63.1 | 53.2 | 63.1 |
| Video-OPD (round 3) | 74.2 | 48.6 | **33.8** | **53.3** | 64.4 | **49.2** | **39.0** | **50.7** | **77.6** | **65.3** | **55.3** | **64.6** |

## D.4. Complete Ablation Results of Multi-Round Video-OPD Training

As illustrated in Table 6, Video-OPD exhibits consistent and monotonic improvements across all three benchmarks as training progresses over multiple rounds. After the first OPD round, the student model already closes a substantial portion of the performance gap to the GRPO-trained teacher, particularly on QVHighlights-TimeLens, where it approaches teacher-level mIoU. A second OPD round further narrows this gap across Charades-TimeLens and ActivityNet-TimeLens, and a third round enables the student to consistently surpass the teacher on all datasets. These results suggest that for temporal video grounding, when the capacity gap between teacher and student is small, a single OPD round is often sufficient for the student to exceed the teacher. In contrast, larger teacher–student scale disparities can be effectively bridged within two to three OPD rounds, highlighting the strong iterative refinement capability and scalability of the proposed Video-OPD framework.

*Table 7.* Comparison of different sampling strategies for the TVDF training curriculum. **Bold** values indicate the best performance.

| Sampling Strategies | Charades-TimeLens | | | | ActivityNet-TimeLens | | | | QVHighlights-TimeLens | | | |
|---|---|---|---|---|---|---|---|---|---|---|---|---|
| | R@0.3 | R@0.5 | R@0.7 | mIoU | R@0.3 | R@0.5 | R@0.7 | mIoU | R@0.3 | R@0.5 | R@0.7 | mIoU |
| DSUS | 69.6 | 40.8 | 27.2 | 48.2 | 58.1 | 42.1 | 33.1 | 45.3 | 70.9 | 55.6 | 45.2 | 56.9 |
| BBDS | 72.3 | 44.3 | 29.1 | 50.3 | 59.5 | 43.9 | 34.3 | 46.5 | 71.1 | 56.0 | 45.2 | 57.1 |
| GWDS | 72.0 | 44.2 | 30.4 | 50.6 | 60.7 | 44.5 | 43.4 | 47.1 | 73.3 | 57.9 | 47.3 | 58.9 |
| Top-$k$ (Ours) | **72.8** | **45.8** | **31.2** | **51.5** | **61.2** | **45.3** | **36.0** | **47.9** | **73.7** | **59.6** | **49.6** | **60.2** |

## D.5. Ablation Study of TVDF Training Curriculum with Different Sampling Strategies

As shown in Table 7, the proposed Top-$k$ sampling strategy consistently achieves the best performance across all three TVG benchmarks and evaluation metrics. Compared with baseline strategies such as DSUS, BBDS, and GWDS, Top-$k$ delivers consistent gains across all metrics, indicating more effective selection of informative training trajectories. These results validate Top-$k$ as a stronger curriculum instantiation for TVDF, leading to more efficient and stable performance improvements than alternative sampling strategies.

*Table 8.* Comparison between Thinking and No-thinking modes of Video-OPD. **Bold** values indicate the best performance.

| Models for Evaluation | Charades-TimeLens | | | | ActivityNet-TimeLens | | | | QVHighlights-TimeLens | | | |
|---|---|---|---|---|---|---|---|---|---|---|---|---|
| | R@0.3 | R@0.5 | R@0.7 | mIoU | R@0.3 | R@0.5 | R@0.7 | mIoU | R@0.3 | R@0.5 | R@0.7 | mIoU |
| Qwen3-VL-8B-Instruct | 61.7 | 41.5 | 23.1 | 42.9 | 41.2 | 30.7 | 20.0 | 30.4 | 46.6 | 38.2 | 29.5 | 36.9 |
| Video-OPD (No-thinking) | **73.1** | **45.8** | **32.4** | **52.0** | **60.5** | **45.6** | **35.8** | **47.3** | **73.8** | **60.3** | **50.4** | **61.0** |
| Video-OPD (Thinking) | 70.3 | 43.1 | 28.7 | 48.7 | 48.5 | 35.2 | 25.6 | 37.0 | 53.9 | 43.4 | 34.5 | 44.1 |

## D.6. Ablation Study of Video-OPD with and without Thinking Mode

Prior work (Li et al., 2025c;b; Patraucean et al., 2023) suggests that for visual perception tasks, reinforcement learning with verifiable rewards (e.g., GRPO) does not benefit from an explicit thinking process. This observation extends to Temporal Video Grounding (TVG), as shown by TimeLens (Zhang et al., 2025). Consistent with these findings, our ablation study in Table 8 shows that incorporating a thinking process during Video-OPD training lowers the Charades-TimeLens mIoU from 52.0 to 48.7 (a 6.3% relative drop). This result confirms that an explicit thinking process likewise provides no advantage for TVG tasks that are primarily driven by visual perception.

## D.7. Prompt Templates Used by Video-OPD Framework during Training and Inference

Figure 7 illustrates the prompt templates used by the Video-OPD framework during training and inference.

---

**Prompt for Video-OPD Framework Training and Evaluation**

**System Instruction:**
You are a video analysis expert.

**User Instruction:**
To accurately pinpoint the event "{query}" in the video, determine the precise time period of the event. Provide the start and end times (in seconds, precise to two decimal places) in the format "start time to end time". For example: "12.54 to 17.83".

---

*Figure 7.* Prompt used for Video-OPD framework training and evaluation.

