# OpenReview forum: "Video-OPD: Efficient Post-Training of Multimodal Large Language Models for Temporal Video Grounding via On-Policy Distillation"
_ICML.cc/2026/Conference — ICML 2026 regular_

### Official Review · Reviewer_X9hQ · 2026-02-20

**Soundness:** 2
**Presentation:** 2
**Significance:** 2
**Originality:** 2
**Overall Recommendation:** 4
**Confidence:** 3

**Summary:**

This paper proposes Video-OPD, a post-training distillation framework that efficiently optimizes temporal video grounding models through on-policy sampling and dense token-level supervision. The overall research is meaningful.

**Compliance With Llm Reviewing Policy:**

Affirmed.

**Final Justification:**

The authors have addressed my concerns; therefore, I maintain my positive recommendation.

**Key Questions For Authors:**

Please see Weaknesses above

I will also take into account the opinions of other reviewers and make a comprehensive decision

**Limitations:**

yes

**Strengths And Weaknesses:**

Strengths:

1. Precise Problem Identification: Clearly pinpoints the core issues of existing methods (SFT, GRPO) in TVG tasks, with well-justified motivation.

2. Solid Experimental Design: Comprehensive comparisons across multiple datasets, training rounds, and different teacher models thoroughly validate the method's generality and robustness.

3. Notable Computational Efficiency: Compared to GRPO, Video-OPD requires only 20% of the training time, offering significant practical value, especially in resource-constrained scenarios.

4. Good Scalability: The method is not dependent on specific model architectures, showing promising potential for transfer and scaling.

Weaknesses:

1. Dependence on Teacher Model Quality: Although experiments show the student can surpass the teacher, the initial teacher still requires a certain level of capability, which might be a limitation in low-resource settings.

2. Lack of Inference Stage Comparison: Although training efficiency is significantly improved, there is no comparison of inference latency or computational overhead, which might be crucial for practical deployment decisions.

---

> ### Author Rebuttal · Authors · 2026-03-30
>
> ## ***Initial Acknowledgment***
> **We are grateful to the reviewers for their valuable and encouraging feedback. The insightful suggestions and comments helped us identify areas for clarification and further improvement. In the following, we address each of the reviewers’ concerns point by point.**
>
> ---
>
> ## ***W1: Dependence on Teacher Model Quality: Although experiments show the student can surpass the teacher, the initial teacher still requires a certain level of capability, which might be a limitation in low-resource settings.***
>
> The discussion in the **Limitations section of the appendix (Section B) may help address this concern** to some extent. Here, we provide a more detailed clarification. As stated in that section, we acknowledge that Video-OPD does rely on a teacher model to provide dense learning signals for updating the policy model. However, **this does not mean that Video-OPD cannot be applied in low-resource settings.**
>
> Importantly, Video-OPD does not rely on an extremely large, general-purpose language model with broad capabilities. In practice, it can instead **perform on-policy distillation from smaller domain-specific expert models** while still achieving strong results. As shown in Table 3 of the paper, the GRPO-trained Qwen3-VL-4B-Instruct serves as a lightweight expert specialized for the temporal video grounding task. **Although this model is even smaller than the policy model, it can still act as an effective teacher** and provide accurate, informative learning signals for policy optimization. Therefore, in practice, Video-OPD may place less demanding requirements on teacher model capacity than one might initially assume.
>
> **We further design a more extreme low-resource setting**. Suppose that for the TVG task, we have **only 2,500 training samples and no effective expert model available**. We first train a Qwen3-VL-4B model on 768 samples to obtain a relatively coarse teacher model. We then use 1,440 samples to train the policy model, Qwen3-VL-8B, with Video-OPD based on this teacher model. Finally, we compare the performance of the resulting model against that of a Qwen3-VL-8B model trained directly with GRPO on the full set of 2,500 samples.
>
> The experimental results are shown below. It can be seen that **even in an extremely low-resource setting, Video-OPD still outperforms GRPO.** This is because we can obtain a coarse teacher model using a smaller model trained on only a very limited amount of data, which in turn helps maintain the effectiveness of Video-OPD in low-resource scenarios.
>
> | Method | Charades | ActivityNet | QVHighlights | Training Samples | GPU Memory | GPU Hours |
> |:------:|:-------:|:-------:|:-------:|:------:|:------:|:-------:|
> | GRPO [Qwen3-VL-8B] | 49.5 | 44.6 | 54.2 | 2500 | 50G | 11.5h |
> | GRPO [Qwen3-VL-4B] | 46.6 | 43.8 | 53.0 | 768 | 27G | 3.6h |
> | Video-OPD [Qwen3-VL-8B] | 50.7 | 47.3 | 59.3 | 1440 | 48G | 0.8h |
>
> More importantly, even in this setting, **Video-OPD still consumes substantially fewer computational resources than GRPO**, even when the cost of training the teacher model is included. As can be seen, the total number of training samples required to first train a teacher model and then apply Video-OPD is still smaller than the number needed for direct GRPO training. At the same time, Video-OPD also incurs lower average GPU memory overhead during training than direct GRPO. Moreover, the total training time of Video-OPD is substantially shorter than that of direct GRPO training.
>
> ---
>
> ## ***W2: Lack of Inference Stage Comparison: Although training efficiency is significantly improved, there is no comparison of inference latency or computational overhead, which might be crucial for practical deployment decisions.***
>
> Below is a comparison of the model’s inference latency and computational overhead at test time after training with Video-OPD and GRPO, respectively, measured in minutes. As shown, the two methods **exhibit almost no difference in inference latency or computational overhead during inference.**
>
> | Method | Charades | ActivityNet | QVHighlights | Video-MME | MVBench | TempCompass |
> |:---:|:---:|:---:|:---:|:---:|:---:|:---:|
> | Baseline | 8.50 | 12.63 | 5.26 | 65.47 | 7.22 | 1.97 |
> | GRPO | 8.60 | 12.12 | 5.67 | 66.16 | 7.05 | 2.17 |
> | Video-OPD | 8.32 | 12.59 | 5.41 | 65.86 | 7.28 | 2.01 |
>
> This is because, in fact, **neither Video-OPD nor GRPO changes the model’s inference paradigm during training**. Therefore, from a theoretical standpoint, the two methods should have almost identical inference latency and computational overhead at test time. For this reason, we did not include an additional comparison of inference latency in the paper.
>
> ---
>
> ## ***Final Acknowledgment***
>
> **We hope our responses have resolved the concerns raised, and we are happy to provide additional clarification if needed. We truly appreciate your thoughtful review, and we wish you all the best in your future research endeavors.**

---

> > ### Author Rebuttal · Reviewer_X9hQ · 2026-04-01
> >
> > Thank you for the authors’ response. My concerns have been addressed, and I will maintain my positive score.

---

> > > ### Author Response · Authors · 2026-04-07
> > >
> > > We sincerely thank the reviewer for the kind follow-up and for taking the time to carefully revisit our paper and rebuttal. We are truly grateful to know that our response has adequately addressed your concerns. We greatly appreciate your thoughtful feedback and engagement throughout the discussion.
> > >
> > > If our clarifications have helped address your concerns and strengthen your overall assessment of the paper, we would greatly appreciate your considering a corresponding increase in the final score. Thank you again for your time, support, and constructive feedback. We also wish you all the best in your future research endeavors.

---

### Official Review · Reviewer_9ef6 · 2026-03-06

**Soundness:** 3
**Presentation:** 3
**Significance:** 3
**Originality:** 3
**Overall Recommendation:** 5
**Confidence:** 4

**Summary:**

The authors propose an efficient on-policy distillation framework for post-training Multimodal Large Language Models (MLLMs) on the Temporal Video Grounding (TVG) task. The method leverages ground truth annotations to select a small set of high-information samples, while employing a teacher model to provide dense token-level rewards. This strategy significantly reduces training cost while achieving superior experimental performance.

**Compliance With Llm Reviewing Policy:**

Affirmed.

**Final Justification:**

I thank the authors for their detailed response. My concerns have been addressed, and I have raised my score accordingly

**Key Questions For Authors:**

Concerns:

1. In the comparison with the state-of-the-art model TimeLens-8B, although Video-OPD ultimately achieves better performance, the current setting uses a maximum input video token length of 8192, which is smaller than the maximum token length (14,336) used in TimeLens. As a result, the performance of Video-OPD is still below the upper-bound performance of TimeLens-8B when it processes longer video token sequences. Given that Video-OPD demonstrates very high training efficiency, it would be interesting to know how much the performance could improve if the number of video tokens in Video-OPD were increased to 14,336.

2. Regarding the finding that learning reasoning chains (chain-of-thought) is not helpful for temporal grounding tasks: although the TimeLens paper reported similar observations, the previous work did not rely on manually annotated reasoning chains. Instead, the model was encouraged to explore a “think-then-answer” strategy autonomously. However, the reasoning steps produced by the teacher model may themselves contain errors. Since there is no ground truth corresponding to the “think” content, the additional reasoning tokens might introduce noise and lead the student model to learn less reliably. Therefore, if high-quality annotated reasoning chains were available, the conclusion that chain-of-thought reasoning is ineffective for perception tasks might not necessarily hold. Furthermore, whether perception tasks require chain-of-thought reasoning may depend on specific data scenarios. For example, in certain long-tail cases—such as when the target object becomes occluded—reasoning could be more important. It is possible that a dual-process system combining fast and slow thinking processes would be more appropriate. The authors may consider discussing this point.

3. The paper shows that reinforcement learning with online distillation can enable the student model to outperform the teacher model. However, is this conclusion specific to the characteristics of the temporal grounding task itself? As shown in Figure 4, in general visual understanding benchmarks, training the model with temporal grounding tasks may actually harm the performance of the base model. Even with the proposed improvements, the performance only matches that of the base model. How can this phenomenon be explained?

**Limitations:**

Yes.

**Strengths And Weaknesses:**

Strengths:

1. The work effectively introduces on-policy distillation into the post-training of video foundation models. Ground truth is cleverly used only as a validation oracle to filter high-quality training samples, greatly improving training efficiency.

2. Experimental results show that the proposed method not only improves training efficiency but also enables the student model to surpass the teacher model after multiple rounds of training.

3. Extensive ablation studies are conducted to validate the effectiveness of the various design choices in the proposed framework.

---

> ### Author Rebuttal · Authors · 2026-03-31
>
> ## ***Initial Acknowledgment***
> **We are grateful to the reviewers for their valuable and encouraging feedback. The insightful suggestions and comments helped us identify areas for clarification and further improvement. In the following, we address each of the reviewers’ concerns point by point.**
>
> ---
>
> ## ***Q1: It would be interesting to know how much the performance could improve if the number of video tokens in Video-OPD were increased to 14,336.***
>
> We used a video token length of 8192 in our paper because most baselines adopt the same setting. To ensure a fair comparison, we standardized the maximum input video token length to 8192 for all methods. When the number of video tokens is increased to 14,336, the performance of Video-OPD is shown below. As can be seen, Video-OPD still significantly outperforms TimeLens-8B.
>
> | Method  | Charades | ActivityNet | QVHighlights |
> |:----:|:----:|:----:|:----:|
> | TimeLens  | 53.9|51.0|64.5|
> | Video-OPD | 55.8| 54.6 |68.3|
>
> ---
>
> ## ***Q2: Discussion on whether reasoning chains are unhelpful for temporal grounding tasks***
>
> This is an excellent question. Since it involves several aspects, we will address it point by point.
>
> ## *1. Is the ineffectiveness of reasoning due to the lack of high-quality annotated reasoning chains?*
>
> There appears to be some misunderstanding about Video-OPD. As shown in Figure 2 of the paper, Video-OPD also requires the policy model to autonomously explore a “think-then-answer” strategy. Therefore, even if high-quality annotated reasoning chains were available, Video-OPD would still not be able to make direct use of them.
>
> ## *2. Is the ineffectiveness of reasoning caused by incorrect reward signals from the teacher model?*
>
> Likely not. Please refer to the results below for GRPO and Video-OPD under the think mode. As can be seen, GRPO also performs worse in the think mode than in the non-think mode. This suggests that the inferior performance of the think mode is not caused by the reward signals provided through teacher guidance.
>
> | Method | Charades | ActivityNet | QVHighlights |
> |:---:|:---:|:---:|:---:|
> | OPD (non-think) | 52.0 | 47.3 | 61.0 |
> | OPD (think) | 48.7 | 37.0 | 44.1 |
> | GRPO (non-think) | 49.5 | 44.6 | 54.2 |
> | GRPO (think) | 46.2 | 35.7 | 42.1 |
>
> ## *3. What exactly causes the model to perform poorly in the think mode?*
>
> We believe this is likely due to the limited reasoning capability of the backbone model. To investigate this, we evaluated the untuned backbone model, Qwen-VL-8B-Instruct, under both think and non-think modes, as shown below.
>
> | Mode | Charades | ActivityNet | QVHighlights |
> |:---:|:---:|:---:|:---:|
> | non-think | 42.8 | 30.4 | 36.9 |
> | think | 36.5 | 18.9 | 16.8 |
>
> The non-think mode performs clearly better than the think mode. This suggests that the backbone model does not inherently possess strong reasoning capabilities. As a result, even after post-training, it is unsurprising that the think mode still underperforms the non-think mode.
>
> ## *4. Does reasoning become more important in long-tail cases?*
>
> Yes. We counted the number of samples on which think mode performs better and those on which non-think mode performs better, as shown below. The results indicate that there are indeed a small number of cases in which think mode yields better answers than non-think mode.
>
> | Mode | Charades | ActivityNet |QVHighlights|
> |:---:|:---:|:---:|:---:|
> | think | 865 | 758 | 284 |
> | non-think | 2498 | 3742 | 1257 |
>
> ---
>
> ## ***Q3: Is the conclusion regarding Video-OPD specific to the characteristics of the temporal grounding task itself? How can the phenomenon shown in Figure 4 be explained?***
>
> ## *1. Is the conclusion specific to TVG task ?*
>
> Certainly not. Please note that this paper is primarily devoted to TVG, and therefore we used **only TVG-related training data** during training. When we replaced the training data with samples from more general video tasks, the results shown below indicate that a similar conclusion also holds for general video reasoning tasks.
>
> | Method | VideoMMMU | LongVideo-Reason | VideoMathQA |
> |:---:|:---:|:---:|:---:|
> | Baseline| 63.3 | 71.5 | 24.3 |
> | Teacher| 64.0 | 78.1| 27.8 |
> | Student| 65.1 | 80.0|30.3|
>
> ## *2. What does Figure 4 illustrate?*
>
> There may be a misunderstanding regarding Figure 4. Because our training was conducted **only** on TVG-related data, our objective was to **verify whether the trained model could retain its generalization ability**, namely, whether it could avoid performance degradation on general video tasks. The results indicate that on-policy methods, such as GRPO and Video-OPD, preserve the model’s generalization ability, whereas off-policy methods, such as OP-RKD, tend to impair it.
>
> ---
>
> ## ***Final Acknowledgment***
>
> **We hope our responses have resolved the concerns raised, and we are happy to provide additional clarification if needed. We truly appreciate your thoughtful review, and we wish you all the best in your future research endeavors.**

---

> > ### Author Rebuttal · Reviewer_9ef6 · 2026-04-06
> >
> > My concerns have been fully addressed. I recommond the authors to add additional experimental results and the discussion on the effectiveness of "think mode".

---

> > > ### Author Response · Authors · 2026-04-07
> > >
> > > Thank you very much for your positive update and for taking the time to revisit our paper. We are truly grateful that you found our responses helpful and that your concerns have been fully addressed.
> > >
> > > We also sincerely appreciate your constructive suggestions regarding the additional experimental results and the more detailed discussion of the effectiveness of think mode, and we will carefully incorporate them into the final version of the paper to further strengthen it.
> > >
> > > Thank you again for your support and thoughtful feedback. We also wish you all the best in your future research endeavors.

---

### Official Review · Reviewer_emm6 · 2026-03-11

**Soundness:** 3
**Presentation:** 3
**Significance:** 2
**Originality:** 2
**Overall Recommendation:** 4
**Confidence:** 4

**Summary:**

This paper introduces a training method to address the issues of sparse supervision and the need for multiple rollouts in GRPO. Specifically, a larger model is used as the teacher to supervise the token production trajectory of the student model. Experimental results demonstrate that it outperforms GRPO on the temporal video grounding task.

**Compliance With Llm Reviewing Policy:**

Affirmed.

**Final Justification:**

In summary, the paper mainly shows through experiments that OPD works well for TVG, which is a useful observation but falls short of delivering a clear technical innovation.

**Key Questions For Authors:**

I think this paper is a new distillation method instead of an RL method. So additional experiments are needed; see details in Weakness.

**Limitations:**

Yes

**Strengths And Weaknesses:**

Strength:

1. This paper is clearly written, and the illustrations are well-designed, making it easy for readers to understand the methodology.
2. The two issues the authors aim to address in applying GRPO to video, i.e., sparse reward signals and the high computational cost caused by multiple rollouts, are indeed important problems.
3. The ablation study is efficient.

Weakness:

1. This paper is essentially a distillation rather than a RL method, using the teacher's knowledge of networks to guide the students' learning of networks.
2. Based on the above, the reported performance gains may not be entirely fair, since other methods (e.g., GRPO) do not leverage knowledge from a larger teacher model.
3. Should also compare the performance with other distillation methods for LLM or MLLMs.
4. Table 1 lacks the performance of SFT.
5. Since the proposed method can also be used for general video understanding, the evaluation on general video benchmarks, e.g., MVBench, VideoMME, NExT-QA and so on, compared with GRPO is also needed.

---

> ### Author Rebuttal · Authors · 2026-03-30
>
> ## ***Initial Acknowledgment***
> **We are grateful to the reviewers for their valuable and encouraging feedback. The insightful suggestions and comments helped us identify areas for clarification and further improvement. In the following, we address each of the reviewers’ concerns point by point.**
>
> ---
>
> ## ***W1: This paper is essentially a distillation rather than a RL method, using the teacher's knowledge of networks to guide the students' learning of networks.***
>
> We respectfully disagree with the characterization of our method as “essentially distillation rather than RL.” Section 3 of [1] provides a detailed argument establishing the fundamental equivalence between RL and on-policy distillation (OPD). Consistent with this view, the current consensus in the on-policy distillation literature is that OPD is properly regarded as a form of RL [2]. Moreover, existing studies on on-policy distillation also consistently adopt standard RL methods, such as GRPO, as baselines [3, 4].
>
> **Importantly, we have already compared it against conventional distillation approaches in our main experiments** to ensure a complete and rigorous empirical evaluation. Please see our responses to ***W2*** and ***W3*** for further details.
>
> ---
>
> ## ***W2&W3: The reported performance gains may not be entirely fair, since other methods do not leverage knowledge from a larger teacher model. Should also compare the performance with other distillation methods for LLM or MLLMs.***
>
> We respectfully note that there seems to be a misunderstanding here. In fact, **our paper already compares Video-OPD against conventional distillation methods**. Please refer to the **third-from-last and second-from-last rows of Table 1: OP-RKD and OP-FKD are precisely the conventional distillation baselines**, and their performance is clearly and consistently inferior to that of Video-OPD. Detailed descriptions of these two distillation methods can be found in Appendix E.
>
> In fact, from the very beginning of our experimental design, we decided to compare Video-OPD **not only with a standard RL baseline (GRPO), but also with conventional distillation methods (OP-FKD, OP-RKD)**, to ensure that the empirical evaluation would be sufficiently thorough and complete.
>
> ---
>
> ## ***W4: Table 1 lacks the performance of SFT.***
> The SFT results are shown below. As can be seen, SFT performs substantially worse than both GRPO and our method.
>
> | Method | Charades | ActivityNet | QVHighlights |
> |:------:|:-----------------:|:--------------------:|:---------------------:|
> | Baseline | 42.1 | 30.6 | 38.1 |
> | GRPO | 48.2 | 44.5 | 54.8 |
> | Video-OPD | 50.4 | 47.3 | 61.5 |
> | SFT | 44.1 | 37.6 | 48.4 |
>
> ---
>
> ## ***W5: Since the proposed method can also be used for general video understanding, the evaluation on general video benchmarks, e.g., MVBench, VideoMME, NExT-QA and so on, compared with GRPO is also needed.***
>
> We respectfully note that there seems to be a misunderstanding here. **Figure 4 in our paper already reports the performance of Video-OPD on general video benchmarks**, including the datasets mentioned by the reviewer, such as MVBench and Video-MME, and also provides direct comparisons with GRPO.
>
> At the same time, we would like to clarify that the primary goal of our paper is to improve performance on the TVG task while **maintaining** generalization to general video understanding tasks. Accordingly, our training data consists exclusively of TVG-related data.
>
> Applying Video-OPD to general video understanding is beyond the scope of the present paper, **but it is entirely feasible**. In practice, simply replacing the training data with data from general video understanding tasks and training with Video-OPD leads to the performance gains shown below.
>
> | Method | VideoMMMU | MMVU | Video-MME | LongVideo-Reason | VideoMathQA |
> |:------:|:-----:|:----:|:----:|:----:|:----:|
> | Baseline | 63.3 | 65.6 | 64.0 | 71.5 | 24.3 |
> | GRPO | 64.0 | 71.6 | 64.5 | 78.1 | 27.8 |
> | Video-OPD | 65.1 | 72.8 | 64.7 | 80.0 | 30.3 |
>
> ---
>
> ## ***Final Acknowledgment***
>
> **We hope our responses have resolved the concerns raised, and we are happy to provide additional clarification if needed. We truly appreciate your thoughtful review, and we wish you all the best in your future research endeavors.**
>
> ---
>
> [1] Yang, Wenkai, et al. "Learning beyond Teacher: Generalized On-Policy Distillation with Reward Extrapolation." arXiv preprint arXiv:2602.12125 (2026).
>
> [2] Lu, Kevin and Thinking Machines Lab, "On-Policy Distillation", Thinking Machines Lab: Connectionism, Oct 2025.
>
> [3] Jin, Woogyeol, et al. "Entropy-Aware On-Policy Distillation of Language Models." arXiv preprint arXiv:2603.07079 (2026).
>
> [4] Zhao, Siyan, et al. "Self-Distilled Reasoner: On-Policy Self-Distillation for Large Language Models." arXiv preprint arXiv:2601.18734 (2026).
>
> [5] Wang, Ye, et al. "Time-r1: Post-training large vision language model for temporal video grounding." arXiv preprint arXiv:2503.13377 (2025).

---

> > ### Author Rebuttal · Reviewer_emm6 · 2026-04-03
> >
> > I would like to thank the authors for their detailed response, which has addressed several of my initial concerns.
> >
> > However, after further investigation, I noticed that the authors did not explicitly acknowledge that On-Policy Distillation (OPD) is a previously proposed algorithm (Lu, K. & Lab, T. M., 2025). The current manuscript does not clearly demonstrate the specific differences between the proposed method and the original OPD. Specifically, it remains unclear whether the authors directly applied OPD to TVG or introduced specific innovations tailored for TVG.
> >
> > Notably, the first two claims of innovation in this paper—namely, the identification of two limitations of GRPO-based post-training and an efficient framework replacing sparse rewards with dense token-level supervision via a reverse KL objective—appear to have been already claimed in the original OPD blog.
> >
> > Therefore, I believe the novelty of this work requires further clarification. I look forward to the authors' explanation of their unique contributions compared to OPD.  At this stage, I would like to maintain my original score until these concerns are adequately addressed.

---

> > > ### Author Response · Authors · 2026-04-04
> > >
> > > First, we are pleased that our rebuttal addressed the concerns raised in the initial review. The reviewer has now raised a completely new concern regarding the novelty of Video-OPD, and we respond to this point below.
> > >
> > > ---
> > >
> > > > ## ***Q1**: I noticed that the authors did not explicitly acknowledge that OPD is a previously proposed algorithm.*
> > >
> > > ***A1***: We respectfully disagree with this statement. In both the Abstract and the Introduction, the two most prominent sections of the paper, we **explicitly state that Video-OPD builds upon the previously proposed OPD**. In addition, we provide **a clear citation** in a highly visible location (bottom-right of Page 1).
> > >
> > > > **Abstract**: *We propose Video-OPD, an efficient post-training framework for TVG **inspired by recent advances in on-policy distillation.***
> > >
> > > > **Introduction**: *We propose Video-OPD, ... **inspired by recent advances in on-policy distillation.***
> > >
> > > This point was also clearly understood by other reviewers. For example:
> > >
> > > > **Reviewer 9ef6**: *The work effectively introduces on-policy distillation into the post-training of video foundation models.*
> > >
> > > ---
> > >
> > > > ## ***Q2**: I believe the novelty of this work requires further clarification.*
> > >
> > > ***A2***: Below, we revisit our contributions and explain where the novelty of the paper lies.
> > >
> > > ### *1. First Introduction of OPD into the Multimodal Domain.*
> > >
> > > To the best of our knowledge, this work is **the first to extend OPD to the multimodal domain**, while also introducing corresponding adaptations to enhance its effectiveness in this new setting (see Point 3). No previous work has investigated whether OPD can be made effective for multimodal post-training, where the learning dynamics and input structure are fundamentally different.
> > >
> > > This point is important: our contribution is not the reinvention of OPD itself, but **the first successful introduction of OPD into a new and substantially more challenging regime**. This is precisely the same type of **contribution that the community has widely recognized** in other settings. For example, **Video-R1 [1] was highly influential not because it invented GRPO, but because it brought GRPO into the multimodal domain** and thereby opened a new research direction in multimodal post-training. **Our paper makes an analogous contribution for OPD**.
> > >
> > > ---
> > >
> > > ### *2. A New Problem Formulation: Why OPD is Uniquely Suitable for TVG.*
> > >
> > > Our work reformulates the solution paradigm for TVG. We do not merely “apply OPD” as a black-box baseline. Rather, we explicitly **identify why TVG is a setting in which OPD is particularly advantageous**, and we articulate this clearly in the **Motivation section**. Specifically:
> > >
> > > - Video inputs are far more expensive than text, making TVG much less tolerant of the overhead from multi-rollout training.
> > >
> > > - TVG also poses a broader, harder exploration problem over dynamic and visually complex content, creating a stronger need for dense token-level supervision.
> > >
> > > **These are not generic observations.** Together, they motivate a new methodological perspective on TVG: OPD is not just applicable, but especially suitable for this problem. This **problem–method alignment** is itself a substantive part of the paper’s novelty.
> > >
> > > ---
> > >
> > > ### *3. Beyond Original OPD: Breaking the Teacher Performance Ceiling.*
> > >
> > > At the technical level, we further propose TVDF, a training curriculum specifically designed for Video-OPD. TVDF allows the model to implicitly exploit supervision signals derived from data labels during training. As a result, Video-OPD and TVDF together form a unified training framework, rather than a simple direct application of original OPD.
> > >
> > > This matters because **it addresses a key limitation of original OPD: the student is typically bounded by the teacher’s performance**. As explicitly acknowledged in the original OPD blog, OPD-trained models consistently remain below their teachers. In contrast, under our jointly designed Video-OPD + TVDF framework, the student is able to clearly outperform the teacher, as shown in Table 3.
> > >
> > > Therefore, even setting aside the significance of being the first to introduce OPD into the multimodal domain, our paper still makes a clear contribution beyond the original OPD: we develop a **training framework that overcomes one of OPD’s most important known limitations**.
> > >
> > > ---
> > >
> > > **TL;DR: Our novelty lies in three aspects:**
> > > - **we are the first to extend OPD from the pure-text domain to the multimodal setting.**
> > > - **we identify and formulate TVG as a problem setting for which OPD is particularly well suited.**
> > > - **we introduce a TVDF curriculum tailored to Video-OPD, yielding a unified framework that overcomes a major limitation of original OPD by enabling the student to surpass the teacher.**
> > >
> > > ---
> > >
> > > ***Acknowledgment: We hope our further clarification helps address the reviewer’s concerns, and we look forward to the reviewer’s further feedback.***
> > >
> > > ---
> > >
> > > [1] Feng, Kaituo, et al. "Video-r1: Reinforcing video reasoning in mllms."

---

### Official Review · Reviewer_VYMW · 2026-03-13

**Soundness:** 3
**Presentation:** 3
**Significance:** 2
**Originality:** 3
**Overall Recommendation:** 5
**Confidence:** 5

**Summary:**

This paper studies post-training for Temporal Video Grounding (TVG) in multimodal LLMs. The key idea is to retain the on-policy property usually associated with RL-based post-training, but replace sparse sequence-level rewards with teacher-provided token-level supervision on student-sampled trajectories. Concretely, the proposed Video-OPD uses reverse-KL distillation on trajectories generated by the current policy, and further introduces Teacher-Validated Disagreement Focusing (TVDF) to prioritize samples that are both teacher-reliable and informative for the student. The paper claims better TVG performance than GRPO with faster convergence and lower computational cost.

**Compliance With Llm Reviewing Policy:**

Affirmed.

**Final Justification:**

I thank the authors for their detailed rebuttal. After carefully considering all of their responses, I believe it is appropriate to raise my score.

**Key Questions For Authors:**

- Can the authors provide a **full compute ledger** comparing Video-OPD and GRPO, including teacher inference cost and total GPU-hours?
- How much of the gain comes from **on-policy sampling** itself versus **reverse-KL dense supervision** versus **TVDF**? A more tightly controlled decomposition would strengthen the causal claim.
- Does TVDF require clean temporal annotations to remain effective, or is it robust under noisier or partially available labels?
- Which parts of the method are expected to transfer beyond TVG to more general video reasoning tasks?

**Limitations:**

yes

**Strengths And Weaknesses:**

### Strengths

- The paper addresses a meaningful problem. The motivation is clear: GRPO-style RL is appealing for TVG because of its on-policy nature, but suffers from sparse rewards and high computational overhead. Recasting the problem as **on-policy distillation with dense token-level supervision** is a sensible and well-motivated direction.
- The proposed method is conceptually clean. Video-OPD preserves alignment between training and inference distributions by sampling from the current policy, while replacing sparse episodic feedback with finer-grained teacher guidance. This is a coherent design choice for temporal grounding.
- The paper appears empirically solid on its target task. According to the abstract and paper framing, the method consistently outperforms GRPO on TVG benchmarks while converging faster. That is a strong practical result if supported by the full experiments.

### Weaknesses

1.  **The efficiency claim is promising but not yet fully closed.**
   The abstract claims “substantially faster convergence and lower computational cost,” but from the paper framing alone it is not yet clear whether this reflects a full end-to-end systems comparison, including teacher-query overhead, or mainly student-side optimization efficiency relative to GRPO. A stronger cost accounting would make the central claim more convincing.

2. **TVDF relies on annotation-informed filtering, which should be described more precisely.**
   The paper states that TVDF prioritizes trajectories that are both teacher-reliable and informative. If reliability is validated against task annotations, then labels are playing a nontrivial role in shaping the effective training distribution. That is reasonable, but the framing should make this dependence explicit.

3. **The broader generalization claim seems stronger than the core evidence.**
   The title and abstract position the paper as an efficient post-training framework for MLLMs, but the method is specifically designed for **temporal video grounding**. The paper would benefit from a more careful discussion of which components are TVG-specific and which are likely to transfer to broader video reasoning settings.

---

> ### Author Rebuttal · Authors · 2026-03-30
>
> ## ***Initial Acknowledgment***
> **We are grateful to the reviewers for their valuable and encouraging feedback. The insightful suggestions and comments helped us identify areas for clarification and further improvement. In the following, we address each of the reviewers’ concerns point by point.**
>
> ---
>
> ## ***W1&Q1: The efficiency claim is promising but not yet fully closed. Can the authors provide a full compute ledger comparing Video-OPD and GRPO, including teacher inference cost and total GPU-hours?***
>
> First, we would like to clarify that the compute cost reported for Video-OPD in the paper **reflects a full end-to-end system-level comparison**, including both the teacher-query overhead and the student-side optimization efficiency. We will make this point more explicit in the revised version of the paper. Here, as requested by the reviewer, we provide a complete compute ledger comparing Video-OPD and GRPO, and the results are shown below.
>
> | Method | Teacher-Query | Total GPU Time |
> |:------:|:------:|:------:|
> | Video-OPD    | 1.09 hour   | 1.9 hour   |
> | GRPO   | ----       | 11.5 hour    |
>
> ---
>
> ## ***W2&Q3: TVDF relies on annotation-informed filtering, which should be described more precisely. Does TVDF require clean temporal annotations to remain effective, or is it robust under noisier or partially available labels?***
>
> A direct answer is that if clean temporal annotations are available, TVDF would perform even better. However, **TVDF can still work effectively even when the labels are noisier or only partially available**.
>
> This is because TVDF consists of two modules: TRPV and DBTP. Among them, only TRPV relies on temporal annotations, while DBTP does not require any temporal annotations at all. **As shown in the ablation study in Table 2 of the paper**, even when only the DBTP module is used, TVDF still brings a 2.3% improvement on the QV-Highlights dataset.
>
> ---
>
> ## ***W3&Q4: Which parts of the method are expected to transfer beyond TVG to more general video reasoning tasks?***
>
> In the motivation section, we explain that Video-OPD is primarily proposed to address two key issues that arise when training MLLMs on TVG tasks with GRPO: (1) sparse rewards lead to ineffective credit assignment, and (2) multi-rollout training incurs prohibitive computational overhead. Therefore, for long-horizon video tasks that require dense learning signals during training and are burdened by the high computational cost of repeated rollouts, **the full Video-OPD framework should be broadly transferable and effective.**
>
> The only limitation lies in the training data. Since this paper is primarily focused on the TVG task, we use **only TVG-related data for training**. To extend the method to more general video reasoning tasks, the training set should be augmented with data from the corresponding target tasks.
>
> **Although this discussion goes somewhat beyond the main scope of the paper**, we nevertheless conducted additional experiments to verify that Video-OPD can be readily transferred beyond TVG to more general video reasoning tasks. Specifically, we additionally sampled examples from general video tasks for training. The results after training are shown below. As can be seen, Video-OPD still significantly outperforms GRPO.
>
> | Method | VideoMMMU | MMVU | Video-MME | LongVideo-Reason | VideoMathQA |
> |:------:|:---------:|:----:|:---------:|:--------:|:------:|
> | Baseline | 63.3 | 65.6 | 64.0 | 71.5 | 24.3 |
> | GRPO | 64.0 | 71.6 | 64.5 | 78.1 | 27.8 |
> | Video-OPD | 65.1 | 72.8 | 64.7 | 80.0 | 30.3 |
>
> ---
>
> ## ***Q2: How much of the gain comes from on-policy sampling itself versus reverse-KL dense supervision versus TVDF? A more tightly controlled decomposition would strengthen the causal claim.***
>
> In fact, the **main experiments and ablation studies in the paper have already quantified the contribution of each of these three components** to the overall performance, albeit in an indirect way. In Table 1, the comparison between Video-OPD and OP-RKD reflects the gain brought by on-policy sampling, while the comparison between Video-OPD and GRPO reflects the gain from reverse-KL dense supervision. In Table 2, the ablation study demonstrates the gain brought by TVDF. We summarize the improvement contributed by each component more explicitly below.
>
> | Dataset | on-policy sampling | dense supervision | TVDF |
> |:---:|:---:|:---:|:---:|
> | Charades | ↑ 4.6% | ↑ 4.8% | ↑ 1.3% |
> | ActivityNet | ↑ 5.2% | ↑ 3.7% | ↑ 2.3% |
> | QVHighlights | ↑ 5.9% | ↑ 8.9% | ↑ 2.7% |
>
> ---
>
> ## ***Final Acknowledgment***
>
> **We hope our responses have resolved the concerns raised, and we are happy to provide additional clarification if needed. We truly appreciate your thoughtful review, and we wish you all the best in your future research endeavors.**

---

> > ### Author Rebuttal · Reviewer_VYMW · 2026-04-04
> >
> > Thanks for the response. All my concerns have been addressed. I'll keep my positive rating score.

---

> > > ### Author Response · Authors · 2026-04-07
> > >
> > > We sincerely thank the reviewer for the kind follow-up and for taking the time to carefully revisit our paper and rebuttal. We are truly grateful to know that our response has adequately addressed your concerns. We greatly appreciate your thoughtful feedback and engagement throughout the discussion.
> > >
> > > If our clarifications have helped address your concerns and strengthen your overall assessment of the paper, we would greatly appreciate your considering a corresponding increase in the final score. Thank you again for your time, support, and constructive feedback. We also wish you all the best in your future research endeavors.

---

### Decision · Program_Chairs · 2026-04-30

**Decision:**

Accept (regular)

**Comment:**

Overall, the paper addresses an important problem, proposes a good method, and provides solid empirical results on temporal video grounding, including improved performance and training efficiency over GRPO. The main concern lies in the novelty: one reviewer argues that the method largely adapts existing on-policy distillation and that the technical contribution beyond OPD is limited, especially regarding the role of TVDF and additional supervision. After the rebuttal, the authors clarified compute cost, added comparisons, and provided additional evidence on generalization and component contributions. I have also read the rebuttal carefully and taken the responses into account. Following the majority of reviewers, I recommend acceptance.